# Gadd45g insufficiency drives the pathogenesis of myeloproliferative neoplasms

Peiwen Zhang[1,2,3,5], Na You[1,2,3,5], Yiyi Ding[1,2,3], Wenqi Zhu[1,2,3], Nan Wang[1,2,3], Yueqiao Xie[1,2,3], Wanling Huang[1,2,3], Qian Ren[1,2,3], Tiejun Qin[1,2,3], Rongfeng Fu[1,2,3], Lei Zhang [1,2,3] ✉, Zhijian Xiao [1,2,3] ✉, Tao Cheng [1,2,3,4] ✉ & Xiaotong Ma [1,2,3] ✉

Despite the identification of driver mutations leading to the initiation of myeloproliferative neoplasms (MPNs), the molecular pathogenesis of MPNs remains incompletely understood. Here, we demonstrate that growth arrest and DNA damage inducible gamma (GADD45g) is expressed at significantly lower levels in patients with MPNs, and *JAK2V617F* mutation and histone deacetylation contribute to its reduced expression. Downregulation of *GADD45g* plays a tumor-promoting role in human MPN cells. *Gadd45g* insufficiency in the murine hematopoietic system alone leads to significantly enhanced growth and self-renewal capacity of myeloid-biased hematopoietic stem cells, and the development of phenotypes resembling MPNs. Mechanistically, the pathogenic role of *GADD45g* insufficiency is mediated through a cascade of activations of RAC2, PAK1 and PI3K-AKT signaling pathways. These data characterize *GADD45g* deficiency as a novel pathogenic factor in MPNs.

Myeloproliferative neoplasms (MPNs) are hematopoietic stem cell (HSC) disorders that affect cells of the myeloid lineage[1]. Classical Philadelphia-negative MPNs (Ph⁻MPNs) are characterized by excessive output of one or more mature myeloid blood cell lineages, and comprise polycythemia vera (PV), essential thrombocythemia (ET) and primary myelofibrosis (PMF)[2–4]. Mutations of *JAK2*, *CALR* and *MPL*, referred to as "driver mutations", occur in more than 95% of MPNs patients, leading to constitutive activation of the JAK-STAT signaling pathway and then the initiation of the diseases[5]. Furthermore, co-occurring somatic mutations in genes associated with epigenetic regulation, transcriptional control and splicing machine have been identified in MPNs patients, preceding driver mutation acquisition and contributing to disease progression. Among them, *ASXL1*, *DNMT3A* and *TET2* are at relatively high frequencies in upwards of 5% of patient samples across the MPNs spectrum[6]. Current treatment has shown

benefits of reduction in splenomegaly and improvements in symptoms in patients with MPNs. However, they cannot eradicate MPNs stem cells efficiently and the molecular response is variable and unpredictable[4,7,8]. A better understanding of the pathogenesis of the MPNs may provide novel targets for developing effective therapeutics.

The GADD45 family of genes (*GADD45a*, *GADD45b* and *GADD45g*), which encode small (18 kDa) nuclear/cytoplasmic proteins, are stress sensors[9]. Our previous work demonstrates that *GADD45g* is preferentially silenced in patients with acute myeloid leukemia (AML) and its upregulation exerts selective and potent anti-leukemic effects[10]. However, whether GADD45g plays a role in the development of MPN remains unknown.

In the present study, we show that *GADD45g* expression is significantly lower in patients with MPNs, and negatively correlated with higher clonogenic potential and inflammatory cytokine production.

[1]State Key Laboratory of Experimental Hematology, National Clinical Research Center for Blood Diseases, Haihe Laboratory of Cell Ecosystem, Institute of Hematology & Blood Diseases Hospital, Chinese Academy of Medical Sciences & Peking Union Medical College, Tianjin, China. [2]Tianjin Institutes of Health Science, Tianjin 301600, China. [3]Center for Stem Cell Medicine, Chinese Academy of Medical Sciences, Tianjin, China. [4]Department of Stem Cell and Regenerative Medicine, Peking Union Medical College, Tianjin, China. [5]These authors contributed equally: Peiwen Zhang, Na You. ✉e-mail: zhanglei1@ihcams.ac.cn; zjxiao@ihcams.ac.cn; chengtao@ihcams.ac.cn; maxt@ihcams.ac.cn

*Gadd45g* deficiency alone is sufficient to cause MPN in mice. GADD45g insufficiency exerts tumor-promoting activities through the activation of RAC2-PAK1-PI3K-AKT signaling pathway. The most common driver mutation *JAK2V617F* and histone deacetylation are involved in the *GADD45g* silencing in MPNs. Our results demonstrate for the first time the pathogenic role of GADD45g insufficiency in MPNs.

## Results

### GADD45g is downregulated in MPNs and its low expression exerts tumor-promoting activities in human MPN cells

In order to assess the role of GADD45g, we first examined the expression levels of this gene in patients with MPNs. The results revealed that its expression was significantly lower in primary CD34$^+$ and bone marrow mononuclear cells (BMMNCs) from patients with ET and PV, compared to those from healthy volunteers, at both mRNA and protein levels (Fig. 1a–c). In a separate cohort (GSE53482), *GADD45g* was markedly downregulated in CD34$^+$ cells from patients with PV and PMF compared to healthy individuals (Fig. 1d). No appropriate datasets of MPNs were available for prognostic analysis.

To explore the functional significance of GADD45g expression, we tested whether the levels of GADD45g in CD34$^+$ cells from patients with MPN were negatively correlated with their colony-forming capacities. We dichotomized these cells into high and low GADD45g expression groups and observed that GADD45g$^{low}$ MPN cells showed a significantly greater clonogenic potential than their respective GADD45g$^{high}$ counterparts (Fig. 1e).

Then, we knocked down GADD45g expression in MPN cell lines HEL and SET-2 by lentiviral delivery of shRNAs (shGADD45g-1 and shGADD45g-2) (Fig. 1f–g). As expected, GADD45g downregulation significantly promoted the clonogenic capacities (Fig. 1h) and proliferation (Fig. 1i), inhibited apoptosis (Fig. 1j), and moderately affected the cell cycle of these cells (Fig. 1k). In contrast, overexpression of GADD45g in the cell lines with a Dox-inducible Tet-on system exerted the opposite effects (Supplementary Fig. 1a–e).

Overall, these findings indicate that GADD45g expression is aberrantly downregulated in MPNs, and its reduced expression acts as a tumor promoter in MPN cells.

### *Gadd45g* deficiency induces aberrations in hematological parameters in mice at 4–6 months of age

We have shown that GADD45g is differentially expressed at low levels in patients with MPN, and cBioPortal analysis revealed that the rates of *GADD45g* gene mutation are extremely low in various myeloid malignancies (<0.5%), including AML, MPNs and MDSs (Supplementary Fig. 2a–c). Therefore, to investigate the role of GADD45g silencing in the pathogenesis of myeloid malignancies, we generated hematopoietic-specific heterozygous (*Gadd45g$^{+/-}$*) and homozygous *Gadd45g* knockout mice (*Gadd45g$^{-/-}$*) by crossing *Gadd45g$^{flox/flox}$* mice with the *Vav-Cre* mice (Supplementary Fig. 3a, b). The expression of the other two members of GADD45 family, *Gadd45a* and *Gadd45b*, was not altered upon *Gadd45g* knockout (Supplementary Fig. 3c).

The hematological parameters and function of long-term HSCs (LT-HSCs) of *Gadd45g*-deleted and Ctrl mice were examined every two months after birth. The results revealed that LT-HSCs from 4-month-old *Gadd45g$^{+/-}$* and *Gadd45g$^{-/-}$* mice showed significantly enhanced self-renewal capacities (Supplementary Fig. 4a, Fig. 2a). No differences in the homing potential were observed (Supplementary Fig. 4b).

Abnormal hematological parameters, including skewed differentiation toward myeloid in the PB, bone marrow (BM) and spleen, as well as evident enlargement of the spleen, were observed in 6-month-old heterozygous and homozygous *Gadd45g* knockout mice (Fig. 2b–h, Supplementary Fig. 5a–f). In addition, both the *Gadd45g$^{+/-}$* and *Gadd45g$^{-/-}$* mice showed dramatic elevations in the number of HSCs in the BM and spleen (Fig. 2i–j). HSC population is

heterogeneous and can be divided into myeloid-biased (My-biased) and lymphoid-biased (Ly-biased) HSC, distinguished by CD150[11,12]. Thus, we next sought to explore whether the myeloid skewing is attribute to a shift in the HSCs pool. Flow cytometric analysis revealed a significant boost in the number of My-biased HSCs in the BM of *Gadd45g$^{+/-}$* and *Gadd45g$^{-/-}$* mice, while the Ly-biased HSCs remained unchanged (Fig. 2k). Correspondingly, the numbers of myeloid-biased MPP2 and MPP3 were markedly augmented, while that of lymphoid-biased MPP4 were unchanged (Fig. 2l). The decreased apoptosis (Fig. 2m), enhanced cell proliferation (Fig. 2n) and increased proportions of cells in S-G2/M phase (Fig. 2o) may account for the augment in the number of My-biased HSCs.

Serial competitive repopulation assays revealed that the self-renewal capacities of My-biased HSCs from 6-month-old *Gadd45g$^{+/-}$* and *Gadd45g$^{-/-}$* mice were significantly enhanced (Fig. 2p), and the differentiation bias toward the myeloid lineage in the BM was reproducible after transplantations (Supplementary Fig. 5g, h). Furthermore, the My-biased HSCs from 8-month-old *Gadd45g*-insufficient mice also exhibited significantly enhanced self-renewal, as compared to age-matched Ctrl mice (Fig. 2q).

Collectively, these data indicate that both heterozygous and homozygous deletions of *Gadd45g* lead to boosts in number and self-renewal capacities of My-biased HSCs, and myeloid-skewed differentiation, gradually evident in mice older than 4 months.

### *Gadd45g* deficiency leads to the development of myeloproliferative neoplasm in mice after 10 months of age

Long-term enhanced self-renewal capacity of HSCs can lead to neoplastic transformation[13,14]. Continuous follow-up showed that *Gadd45g$^{+/-}$* and *Gadd45g$^{-/-}$* mice had a significantly shorter mean survival than Ctrl mice (Fig. 3a). PB, BM and spleen analysis of the moribund mice showed that 25 of 55 *Gadd45g$^{+/-}$* and *Gadd45g$^{-/-}$* mice developed MPN, after 10-month of age, while no Ctrl mice had the disease. There were no significant differences in morbidity or disease types between *Gadd45g$^{+/-}$* and *Gadd45g$^{-/-}$* mice, suggesting the phenotypes induced by *Gadd45g* deficiency are not gene dosage-dependent in mice. Therefore, hereinafter, we no longer discriminated between *Gadd45g* heterozygous and homozygous knockout, but collectively referred to them as "deficiency", or "insufficiency", etc.

All the 25 moribund *Gadd45g*-deficient mice showed anemia and thrombocytosis, and significant increases in the percentages of neutrophils and monocytes, while the proportions of lymphocytes were markedly decreased in the PB, as compared to Ctrl group (Fig. 3b). The increase in platelets was further confirmed by PB smears (Fig. 3c). These mice also exhibited distinct BM hypercellularity, hepatomegaly, and splenomegaly (Fig. 3d–f). Histological analyses demonstrated increased numbers of megakaryocytes in the BM and disorganized spleen architecture, which are hallmarks of MPN (Fig. 3g). Flow cytometric analysis revealed that the proportions of megakaryocytes and myeloid cells were markedly augmented in the BM and spleen (Fig. 3h–k, Supplementary Fig. 6a–d). BM erythropoiesis was reduced (Fig. 3l), with compensatory splenic erythropoiesis (Supplementary Fig. 6e).

Analysis of hematopoietic stem and progenitor cells (HSPCs) revealed that, in the BM of all the moribund *Gadd45g*-insufficient mice, the number of My-biased HSCs was dramatically augmented, while the Ly-biased HSCs remained unchanged (Fig. 2k). The GMP population was markedly increased (Fig. 3m), whereas the number of MEPs was significantly decreased (Fig. 3n). Furthermore, prominent increases in the numbers of My-biased HSCs (Supplementary Fig. 6h), GMPs (Supplementary Fig. 6j) and MEPs (Supplementary Fig. 6l) in the spleens of the moribund mice indicate the occurrence of extramedullary hematopoiesis. These phenotypes are most consistent with a diagnosis of MPN.

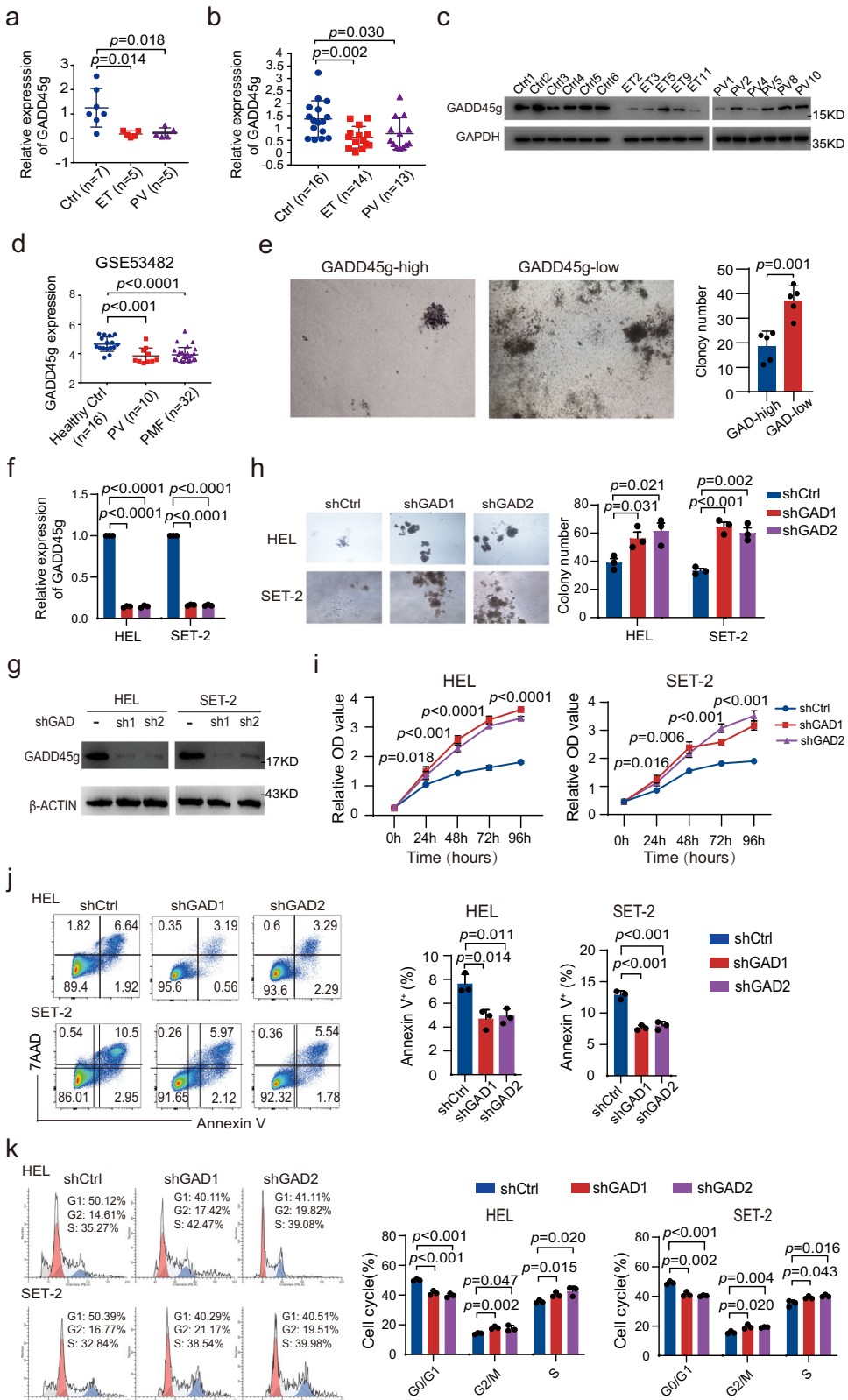

No *Gadd45g*$^{+/-}$ and *Gadd45g*$^{-/-}$ mice developed AML during an observation time of up to 2 years, suggesting that *Gadd45g* depletion is insufficient to engender AML, the development of which requires the acquisition of a second hit.

Noncompetitive transplantations of whole BM cells from the moribund mice with MPN recapitulated the disease in lethally irradiated mice (Fig. 4a–j), with a shortened latency, ranging from 233 to

405 days (Supplementary Table 1), suggesting that the *Gadd45g* deficiency-induced MPN is transplantable.

Next, we generated a Dox-inducible Tet-on lentiviral system to reintroduce *Gadd45g* into the HSPCs isolated from *Gadd45g*-deficient mice with MPN, and transplanted them into lethally irradiated wild-type mice. The reintroduction of *Gadd45g* prominently prolonged the survival and reduced the morbidity of the recipient mice (Fig. 4k,

**Fig. 1 | Reduced expression of GADD45g in human MPNs cells exhibits tumor-promoting functions. a** Relative expression of *GADD45g* mRNA in primary CD34[+] cells from patients with ET (*n* = 5) and PV (*n* = 5) compared with those from human cord blood (CB) (*n* = 7), as determined by qRT-PCR. **b** Relative expression of *GADD45g* mRNA in primary BMMNCs from patients with ET (*n* = 14) and PV (*n* = 13) compared with those from healthy donors (*n* = 16), as determined by qRT-PCR. **c** Western blot analysis of GADD45g protein levels in primary BMMNCs from patients with ET, PV and healthy donors (*n* = 5-6 donors in each group). **d** *GADD45g* expression in CD34[+] cells from patients with PV (*n* = 10) and PMF (*n* = 32) in comparison to those from healthy volunteers (*n* = 16) (GSE53482). **e** Comparison of colony formation potential between primary BM CD34[+] cells from GADD45g[high] and those from GADD45g[low] patients with MPN (*n* = 5 per group). **f** Relative expression of *GADD45g* mRNA in HEL and SET-2 cells transfected with 2 different *GADD45g*-specific shRNAs (shGAD1 and shGAD2). A scrambled shRNA was used as a control (shCtrl). **g** Western blot showing GADD45g-protein levels in HEL and SET-2 cells after knockdown using shGAD1 and shGAD2. Blots are representative of three independent experiments. **h–k** Effects of *GADD45g* knockdown on colony formation (**h**), proliferation (**i**), apoptosis (**j**) and cell cycle (**k**) of HEL and SET-2 cells. For (**f** and **h–k**) Figures shown are representative of three independent experiments with similar results. Data are shown as mean ± SD (*n* = 3 technical replicates). Comparisons were evaluated by two-tailed Student's *t* test, and multiple groups were analyzed with one-way ANOVA.

Supplementary Table 1), confirming the causal role of *Gadd45g* deficiency in the development of MPN in mice.

Somatic mutations are closely associated with MPNs[3]. To determine whether *Gadd45g* deficiency led to occurrence of important malignancy-associated mutations, we performed WES on c-kit[+] BM cells from moribund *Gadd45g*[+/−] mice with MPN, and those from age-matched Ctrl. No apparent differences in the types, rates and spectra of exome mutations were observed between *Gadd45g*-deficient and the Ctrl mice. In addition, no malignancy-associated exome mutations were induced by *Gadd45g* insufficiency (Supplementary Fig. 7a–c).

Taken together, these observations suggest that *Gadd45g* deficiency causes MPN in mice after a period of latency, without inducing additional somatic mutations.

### Gadd45g deficiency induces myeloproliferative neoplasms via activation of PI3K-AKT signaling pathway

We have demonstrated that the median mRNA expression levels of *GADD45g* in BMMNCs from patients with MPNs were approximately twofold lower than in BMMNCs from healthy individuals (Fig. 1b), similar to the reduction in *Gadd45g* haploinsufficient mice (Supplementary Fig. 3b). Therefore, we used *Gadd45g*[+/−] mice in the following experiments. To elucidate the mechanisms underlying the pathogenic role of *Gadd45g* deficiency in MPN, we performed RNA sequencing (RNA-seq) analyses on c-kit[+] BM cells from *Gadd45g*[+/−] mice with MPN in comparison with those from Ctrl (Fig. 5a). Kyoto Encyclopedia of Genes and Genomes (KEGG) enrichment analyses of the upregulated genes revealed that PI3K-AKT signaling pathway was ranked top in *Gadd45g*-deficient cells from mice with the malignancy. Gene set enrichment analysis (GSEA) further confirmed the activation of the pathway (Fig. 5b, c).

The RNA-seq results were validated by Western blot (Fig. 5d, e). Consistent with observations from mouse, knockdown of *GADD45g* in HEL and SET-2 cells resulted in enhanced activities of PI3K and AKT (Fig. 5f, g). These data indicate that *Gadd45g* insufficiency leads to activation of PI3K-AKT signaling pathway.

To determine whether activation of PI3K-AKT pathway plays crucial roles in the hematological abnormalities induced by *Gadd45g* insufficiency, 2-month-old *Gadd45g*[+/−] and Ctrl mice were orally administered with MK-2206, an AKT kinase inhibitor[15], for 4 months. The results revealed that treatment with MK-2206 largely reversed the shortened survival caused by *Gadd45g* deficiency (Fig. 6a). In addition, the boosts in white blood cell (WBC) counts, spleen weight and My-biased HSCs, and the myeloid differentiation bias in the 6-month-old mice were completely abolished (Fig. 6b–h, Supplementary Fig. 8a–h). Consistent with the in vivo results, treatment with MK-2206 completely rescued the enhancements in clonogenicity and proliferation, and the decreases in apoptosis of HEL and SET-2 cells induced by *GADD45g* knockdown (Fig. 6i–k). Moreover, MK-2206 administration largely diminished the clonogenic advantage of primary CD34[+] cells from patients with MPNs in the GADD45g[low] group (Fig. 6l).

These data suggest activation of PI3K-AKT signaling pathway mediates the tumor-promoting role of *Gadd45g* insufficiency in MPN.

### Gadd45g deficiency induces the activation of PI3K-AKT pathway via RAC2-PAK1 axis

All GADD45 family members are small, acidic proteins, and often function through interactions with other regulatory proteins[16]. To identify possible proteins that directly interact with GADD45g, we performed GADD45g-interactome screening using immunoprecipitation (IP) followed by mass spectrometry (MS) in c-kit[+] BM cells of wild-type mice, and identified RAC2, the only protein expressed exclusively in hematopoietic cells and promoting the progression of MPNs[17], as a potential interacting protein (Supplementary Data 1). The direct interaction of GADD45g with RAC2 was validated using co-immunoprecipitation (co-IP) assays followed by Western blot (Fig. 7a), and was further confirmed by the evident colocalization of GADD45g with RAC2 (Fig. 7b). Meanwhile, RAC1, which exhibits considerable homology and is functionally interchangeable with RAC2[18], showed no interaction with GADD45g (Supplementary Fig. 9a, b). Next, we examined the cellular localization of the three proteins by immunofluorescence analysis of cord blood CD34[+] cells from healthy human donors. Both GADD45g and RAC2 exhibited a perinuclear distribution, and areas of colocalization of them were clearly evident, consistent with that visualized in murine c-kit[+] BM cells; whereas RAC1 displayed dispersed cellular locations with little perinuclear localization and its colocalization with GADD45g was low (Fig. 7c). Therefore, the distinct cellular localizations of RAC2 and RAC1 may be implicated in the specificity of binding.

Intriguingly, the interaction of RAC2 with GADD45g was prominently reduced, and the level of activated RAC2-GTP was significantly increased upon heterogeneous loss of *Gadd45g* (Fig. 7a, d). Consistent results were obtained in *GADD45g*-knocked down human MPN cell lines (Fig. 7e, f). In line with these data, our RNA-seq analyses also revealed activation of Rho GTPases, a family to which RAC2 belongs, in c-kit[+] BM cells from moribund *Gadd45g*[+/−] mice (Supplementary Fig. 10). Collectively, these data suggest that GADD45g interacts with and suppresses the activity of RAC2 in wild-type cells, and *Gadd45g* insufficiency leads to the disassociation of GADD45g-RAC2 complex and relieves the inhibition of RAC2 activation.

PAKs are serine/threonine kinases which act as major downstream effectors of the Rho-GTPases family[19–21]. Among the PAK family members, only activation of PAK1 has been shown to be implicated in the pathogenesis of MPNs[22]. In addition, Lu et al. reported that activated RAC2 regulates homing of T lymphoid progenitor through successive activations of PAK1 and AKT[23]. These observations led us to speculate that activated RAC2 phosphorylated PAK1, which in turn resulted in activation of PI3K-AKT.

To test the hypothesis, we first detected the activation status of PAK1 in c-kit[+] BM cells from moribund *Gadd45g*[+/−] and Ctrl mice. As expected, *Gadd45g* deficiency resulted in significant enhancement of PAK1 activity (Fig. 7g). Consistent results were obtained in *GADD45g*-knocked down HEL and SET-2 cells (Fig. 7h).

We next investigate whether the activation of PI3K-AKT induced by *GADD45g* knockdown is mediated through RAC2-PAK1 pathway in MPN cell lines. As expected, downregulation of RAC2 by lentiviral

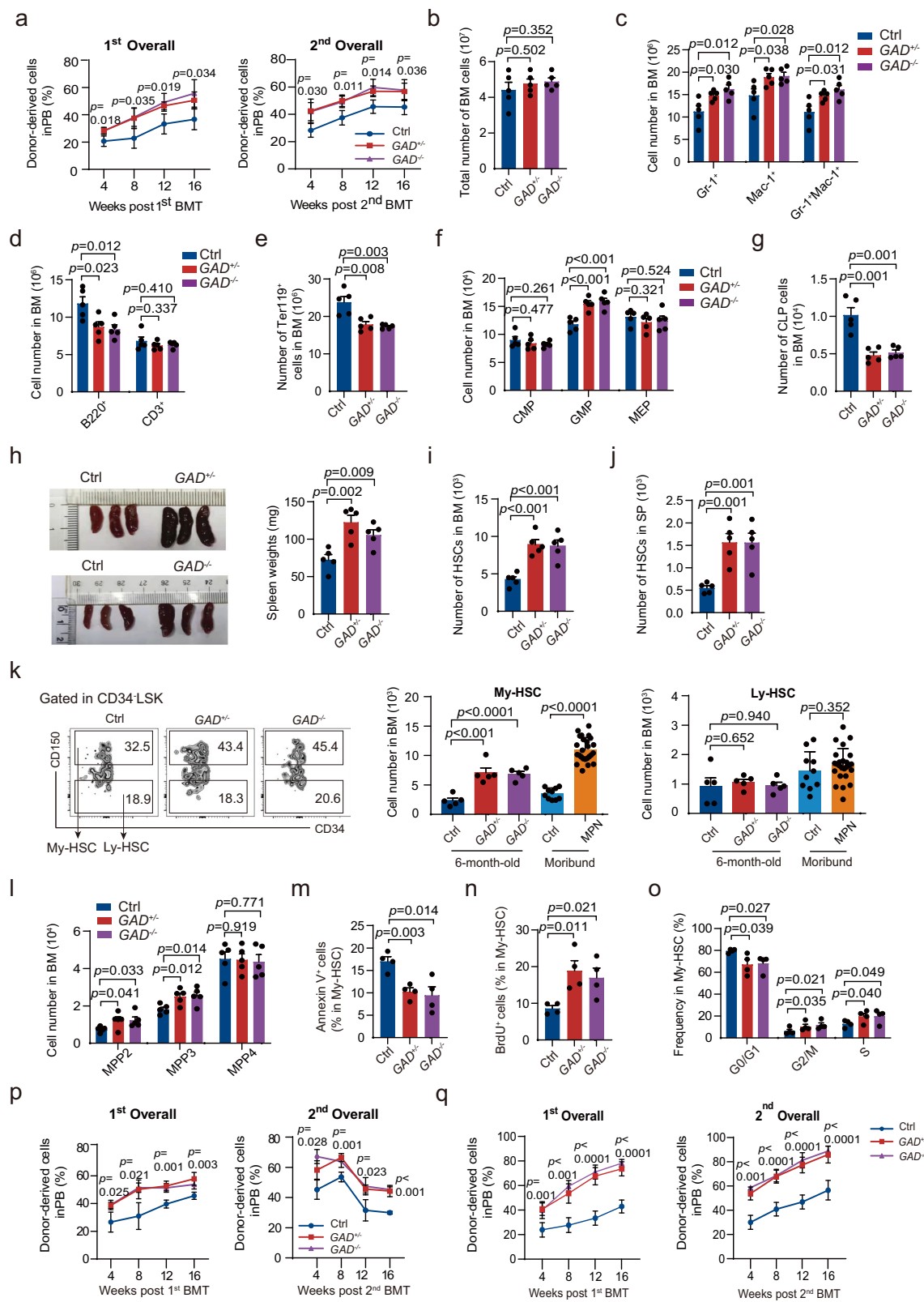

shRNA not only largely diminished the activation of PAK1 and PI3K-AKT (Supplementary Fig. 11a, b, Fig. 7i), but also reversed the enhancement of colony-forming capacity and inhibition of apoptosis caused by *GADD45g* knockdown in HEL and SET-2 cells (Fig. 8a, b). Administration of IPA-3, a specific inhibitor of PAK1[24], exerted similar effects on PI3K-AKT (Fig. 7j), colony formation and apoptosis in these cells (Fig. 8c, d).

Similar to AKT kinase inhibitor MK-2206, treatment with RAC inhibitor EHT 1864[25] or PAK1 inhibitor IPA-3 largely diminished the clonogenic advantage of CD34+ cells from MPNs patients in the GADD45g[low] group (Fig. 8e, f).

Collectively, our study demonstrates that the induction of MPN by *Gadd45g* insufficiency involves a cascade of signaling events, including successive activations of RAC2, PAK1 and PI3K-AKT.

**Fig. 2 | *Gadd45g* deficiency leads to aberrations of hematological parameters and enhanced self-renewal capacities of HSCs in mice aged 4–6 months.**
**a** Percentage of donor chimerism in the PB of lethally irradiated primary recipients transplanted with LT-HSCs freshly isolated from 4-month-old *Gadd45g*[+/-] (*GAD*[+/-]) (*n* = 4), *Gadd45g*[-/-] (*n* = 4) and Ctrl (*n* = 5) mice together with competitor cells (top). Donor chimerism in the PB of secondary recipients transplanted with CD45.2[+] LT-HSCs from primary recipient mice together with competitor cells (bottom).
**b–g** Total number of BM cells (**b**), and absolute numbers of Gr-1[+], Mac-1[+] and Gr-1[+]Mac-1[+] cells (**c**), B220[+] and CD3[+] cells (**d**), Ter119[+] cells (**e**), CMP, GMP, MEP cells (**f**), and CLP cells (**g**) in the BM of 6-month-old *Gadd45g*[+/-], *Gadd45g*[-/-] and Ctrl mice (*n* = 5 mice per group). **h** Representative images of spleens and spleen weights of 6-month-old *Gadd45g*[+/-], *Gadd45g*[-/-] and Ctrl mice (*n* = 5 mice per group). **i, j** Absolute numbers of HSCs in the BM (**i**) and spleen (**j**) of 6-month-old *Gadd45g*[+/-], *Gadd45g*[-/-] and Ctrl mice (*n* = 5 mice per group). **k** Numbers of My-biased HSCs and Ly-biased HSCs in the BM from 6-month-old *Gadd45g*[+/-], *Gadd45g*[-/-] and Ctrl mice (*n* = 5 mice per group), and from the moribund mice with MPN (*n* = 25) and Ctrl (*n* = 10) group

(bottom). Representative flow cytometry plots were shown on the top. **l** Absolute numbers of MPP2, MPP3 and MPP4 in the BM of 6-month-old *Gadd45g*[+/-], *Gadd45g*[-/-] and Ctrl mice (*n* = 5 mice per group). **m–o** Apoptosis (**m**), proliferation (**n**) and cell-cycle (**o**) analysis of My-biased HSCs in the BM from 6-month-old *Gadd45g*[+/-], *Gadd45g*[-/-] and Ctrl mice (*n* = 4 mice per group). **p** Percentages of donor-derived overall (CD45.2[+]) cells in the PB of primary recipients transplanted with My-biased HSCs from 6-month-old *Gadd45g*[+/-] (*n* = 4), *Gadd45g*[-/-] (*n* = 4) and Ctrl (*n* = 5) mice together with competitor cells (top), and chimerism in the PB of secondary recipients transplanted with CD45.2[+] My-biased HSCs from primary recipients together with competitor cells (bottom). **q** Percentages of donor chimerism in the PB of primary recipients transplanted with My-biased HSCs from 8-month-old *Gadd45g*[+/-], *Gadd45g*[-/-] and Ctrl mice together with competitor cells (left), and in the PB of secondary recipients transplanted with CD45.2[+] My-biased HSCs from primary recipients together with competitor cells (right). (*n* = 5 mice per group). For (**a–q**): Data are shown as means ± SD, two-tailed Student's *t*-test.

## *Gadd45g* deficiency induces aberrant inflammatory cytokine production

It is well accepted that chronic inflammation plays a critical role in the pathogenesis of MPNs[26]. To investigate whether *Gadd45g* insufficiency led to a dysregulated cytokine production, we detected the serum concentrations of several inflammatory cytokines, which have been reported to be of pivotal significance in MPNs[27-33], using Luminex assay in the 4- and 6-month-old and moribund *Gadd45g*[+/-] mice, and their age-matched controls. Intriguingly, the results revealed that the levels of IL-4, IL-6, CCL3 and GM-CSF were significantly increased in *Gadd45g*[+/-] mice after the onset of MPN (Fig. 9a), consistent with clinical observations from the patients[27,32,33].

In addition, to examine whether the levels of GADD45g were negatively correlated with the cytokine production, some of the patients with MPNs included in our GADD45g expression analysis were conducted simultaneously serum cytokine measurement. We dichotomized these patients into GADD45g[high] and GADD45g[low] expressers and observed that the levels of IL-4 and IL-6 were significantly higher in GADD45g[low] patients with MPNs, as compared with their respective GADD45g[high] counterparts (Fig. 9b).

## *JAK2V617F* mutation and histone deacetylation are involved in *GADD45g* silencing in MPNs, and *GADD45g* downregulation partially mediates *JAK2V617F* activity in a MPN xenograft model

Our previous study reveals that FLT3-ITD and MLL-AF9 oncogenes, which present with a high leukemic burden and confer a poor prognosis in patients with AML, contributed to the silencing of *GADD45g* in AML[10]. These findings urged us to investigate whether *JAK2V617F*, the most recurrent mutation critical for disease initiation, is involved in the downregulation of *GADD45g* in MPNs. We treated primary BMMNCs from patients with PV, 95% of which harbor the *JAK2V617F* mutation[3], with JAK2 inhibitor (ruxolitinib) or vehicle. qRT-PCR analysis showed that the mRNA expression levels of *GADD45g* were markedly increased upon ruxolitinib treatment (Fig. 10a). We next transduced 293T cells with lentiviruses expressing *JAK2V617F*, wild-type *JAK2*, or empty vector constructs, and observed that only *JAK2V617F* overexpression resulted in a significant decrease in the expression level of *GADD45g* (Fig. 10b). These data suggest that *JAK2V617F* mutation can repress GADD45g expression.

We then used a xenograft model to assess the involvement of *GADD45g* reduction in the pathogenesis of *JAK2V617F* MPN in vivo. Luciferase-expressing HEL92.1.7 cells (HEL92.1.7-Luc) were transfected with Dox-inducible *GADD45g*-specific shRNA and engrafted into NSG mice by intravenous injection. The engraftment could be detected in vivo 10 days after transplantation, as indicated by the bioluminescence signal (Fig. 10c). The mice were then treated with either ruxolitinib or vehicle. Ruxolitinib administration resulted in a significant reduction in tumor burden (Fig. 10d), and a prominent elevation of

*GADD45g* expression in hCD45-positive BMMNCs (Fig. 10e), which is consistent with our in vitro observations (Fig. 10a). Next, to determine whether downregulation of *GADD45g* had a restorative effect on disease burden, each of the ruxolitinib and vehicle treatment groups was subdivided into Dox and non-Dox treatment groups. The results revealed that, as expected, treatment with Dox resulted in an apparent increase in tumor burden and a significant decrease in survival compared with the vehicle (Fig. 10f, g). Furthermore, Dox-induced *GADD45g* knockdown with concurrent *JAK2* inhibition led to a partial restoration of tumor burden and a shortened survival, as compared to treatment with ruxolitinib alone (Fig. 10f, g). These observations indicate GADD45g partially mediates the pathogenicity of *JAK2V617F* in the MPN xenograft model.

Our previous work has shown that in addition to oncogenes, both promoter DNA methylation and histone deacetylation are also associated with *GADD45g* inactivation in AML[10]. To investigate whether DNA methylation is involved in the decreased expression of *GADD45g* in MPNs, we treated MPN cell lines HEL and SET-2, and primary BMMNCs from MPN patients with DNA demethylating agent Decitabine. The mRNA level of *GADD45g* was upregulated by about twofold in these cells (Fig. 10h, i). Our methylation-specific PCR (MS-PCR) assay revealed that methylated GADD45g alleles were significantly decreased in HEL cell line while remained unchanged in SET-2 cell line upon Decitabine treatment (Fig. 10j). These data suggest promoter DNA methylation plays a marginal role in the silencing of *GADD45g*.

To explore the contribution of histone modifications to the reduced expression of *GADD45g* in MPNs, we analyzed gene expression profilings of CD34[+] BMMCs/PBMCs from MPN patients and found that *GADD45g* showed a marked inverse correlation with *HDAC1* and *HDAC2* expression (Fig. 10k–l). Our previous study has shown that the selective HDAC1/2 inhibitor Romidepsin exhibits the strongest induction of *GADD45g* in AML[10]. We then treated HEL and SET-2 cells and BMMNCs from MPN patients with Romidepsin, and observed that the mRNA levels of *GADD45g* were upregulated more than 4-fold in the cell lines (Fig. 10m), and approximately 3- to 4-fold in primary BMMNCs (Fig. 10n). ChIP-qPCR assay revealed that the levels of histones H3 and H4 acetylation at the GADD45g promoter were significantly upregulated upon Romidepsin treatment in both HEL and SET-2 cell lines (Fig. 10o, p).

Together, we show that the reduced expression of *GADD45g* in MPNs is associated with *JAK2V617F* mutation and histone deacetylation, and *GADD45g* reduction partially mediates the pathogenic role of *JAK2V617F* mutation in MPN cells.

## Discussion

MPNs are chronic myeloid neoplasms arise from a somatic mutation in the pluripotent hematopoietic stem cell[4,34]. However, current treatment for MPNs is only symptomatic and has modest benefits[4],

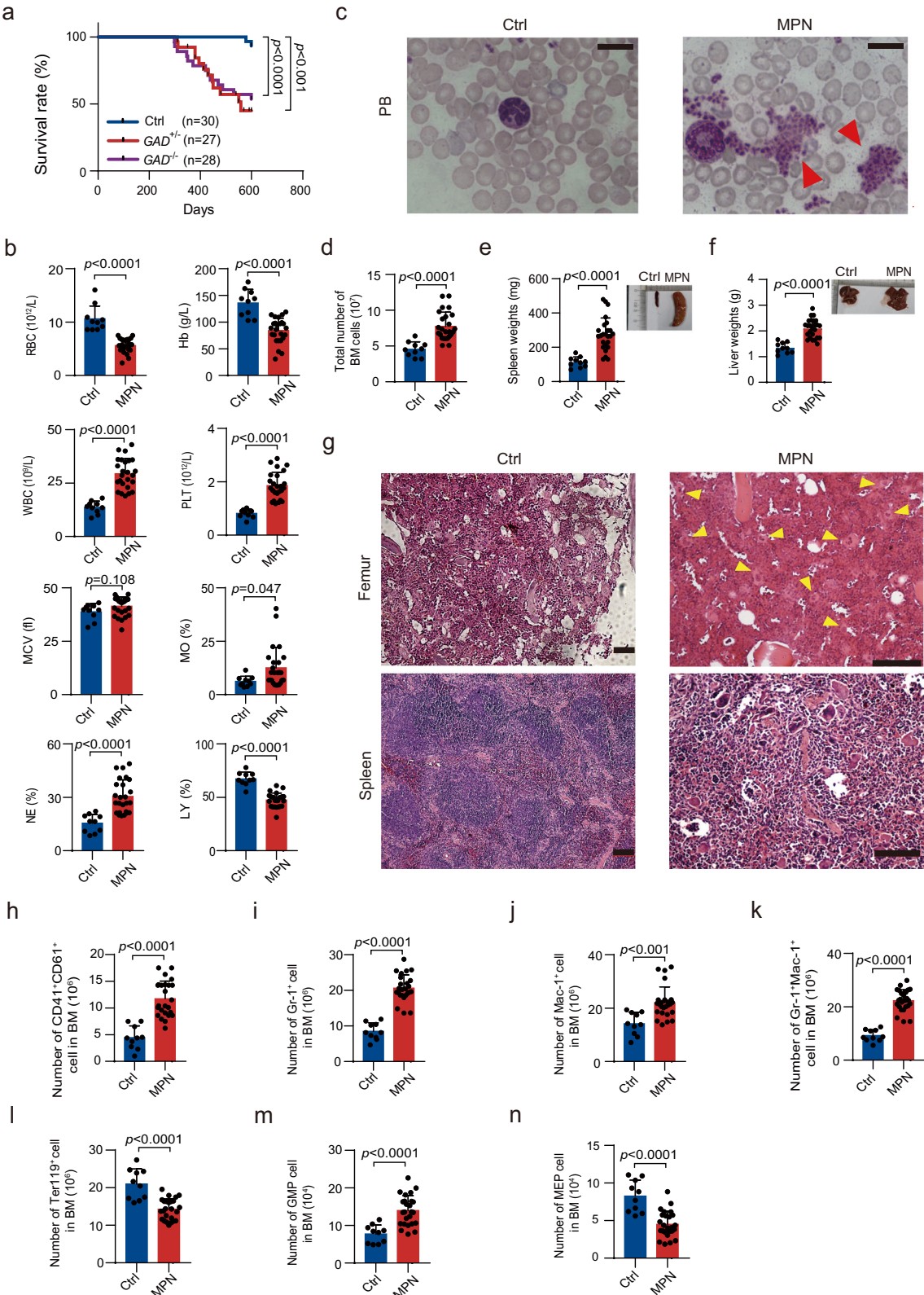

highlighting the need to further understand the pathogenesis of MPNs. Here, we identify GADD45g deficiency as a novel pathogenic factor in MPNs.

MPNs are clonal disorders originating from HSCs[34]. In our mouse model, increased self-renewal capacities of HSCs was the only hematopoietic change in *Gadd45g*[+/−] and *Gadd45g*[−/−] mice as early as 4 months of age. Thereafter, the enhancement in number and function

of HSCs and myeloid lineage skewing became gradually evident over time, resulting in malignant transformation eventually. These data indicate that the myeloid malignancies engendered by *Gadd45g* deficiency may arise in the HSCs compartment. Intriguingly, analysis of HSC heterogeneity reveals that the increments in number and self-renewal capability only occurred in My-biased HSCs but not Ly-biased HSCs before and after the onset of the diseases, implicating that My-

**Fig. 3 | *Gadd45g* deletions induce MPN in mice after 10 months of age. a** Kaplan-Meier survival curves of *Gadd45g*[+/-], *Gadd45g*[-/-] and Ctrl mice (*n* = 27-30 mice per group, log-rank test). **b** Counts of RBC, hemoglobin, WBC, platelet, and MCV, and percentages of monocytes, neutrophils and lymphocytes in the PB of moribund *Gadd45g*-insufficient mice with MPN (*n* = 25) and Ctrl mice (*n* = 10). **c** Wright-Giemsa staining of PB smears prepared from Ctrl and mice with MPN. Red arrowheads indicate abundant platelets. Bar represents 50 μm. At least three independent biological replicates were performed. **d** Total number of BM cells in moribund mice with MPN (*n* = 25) and Ctrl mice (*n* = 10). **e** Spleen weights and representative images of spleens of moribund mice with MPN (*n* = 25) and Ctrl mice (*n* = 10). **f** Liver weights and representative images of livers of moribund mice with MPN (*n* = 25) and Ctrl mice (*n* = 10). **g** H&E staining of femur (top) and spleen (bottom) sections from Ctrl and moribund mice with MPN. Yellow arrowheads indicate mega-karyocytes. Bars represent 100 μm. At least three independent biological replicates were performed. **h–n** Absolute number of CD41[+]CD61[+] (**h**), Gr-1[+](**i**), Mac-1[+](**j**), Gr-1[+]Mac-1[+] (**k**), Ter119[+] (**l**), GMP (**m**), MEP (**n**) cells in the BM of moribund mice with MPN (*n* = 25) and Ctrl mice (*n* = 10). For (**b**, **d–f**, **h–n**): Data are shown as means ± SD. Two-tailed Student's *t* test.

biased HSCs may serve as the cellular basis of the tumor development in mice. Whether My-biased HSCs are also augmented in human patients is worthy of further investigations.

Genetic and epigenetic alterations are known to play a key role in silencing of tumor suppressor genes[34,35]. Our previous study reveals that *GADD45g* is repressed by oncogenic drivers FLT3-ITD and MLL-AF9, promoter DNA methylation and histone deacetylation in AML[10]. In the present study, we observed that the reduced expression of *GADD45g* in MPNs is attributed to *JAK2V617F* mutation, the most common driver mutation in MPNs, and histone deacetylation; while the impact of DNA methylation is modest. The finding that the *JAK2V617F* mutation is associated with aberrant expression of GADD45g is of interest. Our previous work has shown that NF-κB binds directly to *GADD45g* promoter and represses *GADD45g* transcription in AML cells[10]. Scuto et al. reported that STAT3 interacts with NF-κB to co-repress the expression of *GADD45g* in malignant lymphoid cells[36]. In addition, c-Myc suppresses *GADD45g* gene expression in prostate cancer cells[37,38]. Notably, *JAK2V617F* mutation is reported to induce the activation of NF-κB and STAT3[3,39], as well as constitutive expression of c-Myc[40] in MPNs. Therefore, activations of NF-κB, STAT3 and c-Myc, either singly or in combination, might contribute to the down-regulation of *GADD45g* by *JAK2V617F* mutation in MPNs, which warrants further study. Nevertheless, it is worth noting that *JAK2V617F* mutation is not the only regulatory mechanism responsible for the low expression of *GADD45g* in MPNs, and that GADD45g reduction only partially mediates the tumor-promoting activities of *JAK2V617F* muta-tion in a xenograft model of MPN. Furthermore, *Gadd45g* deficiency alone is sufficient to induce MPN in mouse. These data support GADD45g as a novel pathogenic mechanism in MPNs.

Our data demonstrate that the pathogenic role of *GADD45g* insufficiency in MPN is mediated through successive activations of RAC2, PAK1, and PI3K-AKT in mice and human MPN cell lines. Although constitutive activation of JAK-STAT signaling is considered critical in the development of MPNs, aberrant activations of RAC2, PAK1 and PI3K-AKT have been separately described[17,22,41,42]. Inhibitors of PI3K-AKT have been proved to exert anti-tumor effects in both mouse models and cell lines of the disease[43,44]. Our data further highlight the potential roles of these signalings in the pathogenesis of MPNs and suggest possible therapeutic targets for the treatment of the diseases. However, it should be pointed out that the establishment of the rele-vance of the level of *GADD45g* to the activation of RAC2-PAK1-PI3K-AKT axis in patients with MPNs was impeded by a lack of adequate primary cells, and further studies are needed to clarify such a relevance.

In general, gene knockout in mice exhibits a dosage-dependent manner, and loss of one allele often has subtle effects[45]. In the present study, we find that the morbidity and disease types engendered by heterozygous and homozygous deletions of *Gadd45g* are similar. One possible explanation is that the activated RAC2 released due to a 50% reduction in GADD45g protein is sufficient to induce constitutive activations of downstream signalings, which in turn lead to develop-ment of the disease.

Chronic inflammation is a hallmark feature of MPNs. The elevated levels of pro-inflammatory cytokines are considered to be upregulated and sustained by somatic mutations[27,28]. Intriguingly, we observed that murine *Gadd45g*[+/-] MPN share some cytokine characteristics with patients with the disease. Furthermore, lower expression of *GADD45g* correlates with higher concentrations of IL-4 and IL-6 in patients with MPNs. Therefore, *GADD45g* inactivation may act as the trigger for the aberrant production of some inflammatory cytokines. Whether *GADD45g* inactivation represents an additional mechanism or med-iates the increase in these factors induced by somatic mutations needs further studies and large cohort analysis.

Collectively, this study for the first time characterizes GADD45g insufficiency as a novel pathogenic factor in MPNs, functioning through activation of the RAC2-PAK1-PI3K-AKT cascades. The low expression of *GADD45g* in MPNs is attributed to *JAK2V617F* mutation and histone deacetylation. Our *Gadd45g* deletion mouse model pro-vides a valuable platform for not only unveiling the molecular basis for the initiation and progression of the disease, but also developing novel therapies for the patients.

## Methods

### Ethics statement
All animal experiments in this study were performed in compliance with institutional guidelines and approved by the Institutional Animal Care and Use Committees of State Key Laboratory of Experimental Hematology (SKLEH). All laboratory experiments with primary sam-ples were conducted according to the ethical principles for medical research and approved by the Ethics Committee of the Institute of Hematology and Blood Diseases Hospital.

### Mice
*Gadd45g*[flox/WT] mice (6-8 weeks old, 5 females, C57BL/6 N strain) were purchased from Biocytogen Pharmaceuticals (Beijing, China). C57BL/6 J and C57B6.SJL mice (6-8 weeks old, male and female) were pur-chased from the animal facility of the State Key Laboratory of Experi-mental Hematology (SKLEH). NOD-*Prkdc*[scid]*IL2rg*[tm1](NSG) mice (6-8 weeks old, 40 females, strain# N-000002) were purchased from Beijing HFK Bio-Technology Co. Ltd (Beijing, China). B6.Cg-*Commd10*[Tg(Vav1-icre)A2Kio]/J (*Vav-Cre*) mice (6-8 weeks old, 2 males and 2 females, strain# 008610) were purchased from Jackson Lab. *Gadd45g*[flox/WT] mice were crossed with *Vav-Cre* mice to generate *Gadd45g*[flox/WT]; *Vav-Cre* and *Gadd45g*[flox/flox]; *Vav-Cre* mice. Mice were maintained at macroenvironmental temperature of 21–22 °C, humidity (48–52%), in a conventional 12:12 light/dark cycle with lights on at 6:00 a.m. and off at 6:00 p.m. All animal studies were approved by the Institutional Animal Care and Use Committees of SKLEH. All mice were kept under specific pathogen-free conditions with free access to food and water in accordance to Swiss federal regulations. Both male and female mice were used in all the studies. To determine the earliest timepoint at which abnormal BM hematopoiesis was induced by *Gadd45g* deficiency, mice were euthanized via cervical dislocation every two months for the assessment of hematological parameters. Mice older than 10 months of age were euthanized when they exhib-ited moribund characteristics.

### Cell culture
HEL (TIB-180) and SET-2 (ACC 608) cells were obtained from American Type Culture Collection (ATCC) and German Collection of

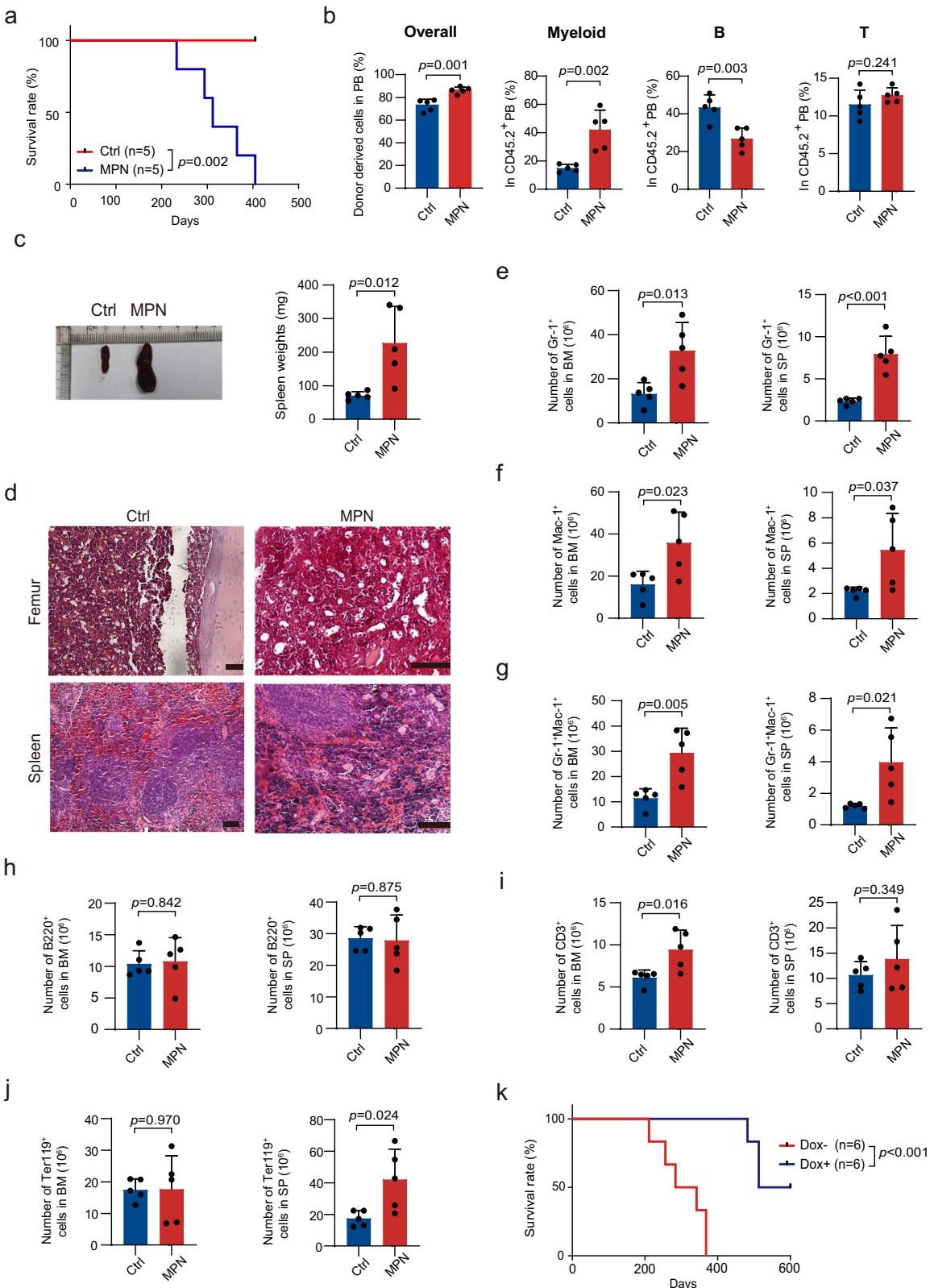

Microorganisms and Cell Cultures, respectively. Cells were cultured in RPMI 1640 supplemented with 10% fetal bovine serum (FBS). Luciferase-expressing HEL92.1.7 cells (HEL92.1.7-Luc cells, NM-B24-TG01) were purchased from Shanghai Model Organisms Center, Inc. (Shanghai, China), and cultured in RPMI 1640 supplemented with 10% FBS and 6 µg/ml Blasticidin S HCl (ST018, Beyotime, Shanghai, China).

293T cells (CRL-11268) were obtained from ATCC and cultured in Dulbecco's modified Eagle's medium supplemented with 10% FBS. Cells were grown at 37 °C in a humidified atmosphere containing 5% $CO_2$. HEL, SET-2 and 293T cell lines were validated using STR analysis by the vendors. Luciferase activity assay was performed before inoculating HEL92.1.7-Luc cells by the vendors.

**Fig. 4 | *Gadd45g* deficiency-induced MPN is transplantable and reintroduction of *Gadd45g* prominently prolongs the survival of mice with MPN. a–j** One million whole BM cells from the Ctrl mice or moribund mice with MPN (CD45.2) were transplanted into lethally irradiated recipients (CD45.1) (5 recipients per group). The hematological parameters were examined when the recipient mice required euthanasia because of moribund conditions. Kaplan-Meier survival curves of the recipient mice (log-rank test) (**a**). Percentages of donor-derived overall (CD45.2$^+$) and those of myeloid (Mac-1$^+$), B (B220$^+$), and T (CD3$^+$) cells in CD45.2$^+$ PB of recipients (**b**). Representative images of spleens from recipient mice (left) and spleen weights of recipient mice (right) (**c**). H&E staining of femur (top) and spleen

(bottom) sections from recipient mice. Bars represent 100 μm. At least three independent biological replicates were performed (**d**). Absolute number of Gr-1$^+$ (**e**), Mac-1$^+$ (**f**), Gr-1$^+$Mac-1$^+$ (**g**), B220$^+$ (**h**), CD3$^+$ (**i**) and Ter119$^+$ (**j**) cells in the BM (left) and spleen (right) of recipient mice. (**k**) Lin⁻c-kit$^+$ HSPCs isolated from moribund mice with MPN were transfected with Dox-inducible *Gadd45g* lentiviral vector and then transplanted into lethally irradiated recipient mice (10$^4$ cells per mouse). Dox (1 mg/mL) was added to drinking water 7 days after transplantation for 1 week to induce *Gadd45*g expression. Kaplan-Meier survival curves of recipient mice in different groups were shown (*n* = 6 mice per group, log-rank test). For (**b**, **c**, **e**–**j**): Data are shown as means ± SD. Two-tailed Student's *t* test.

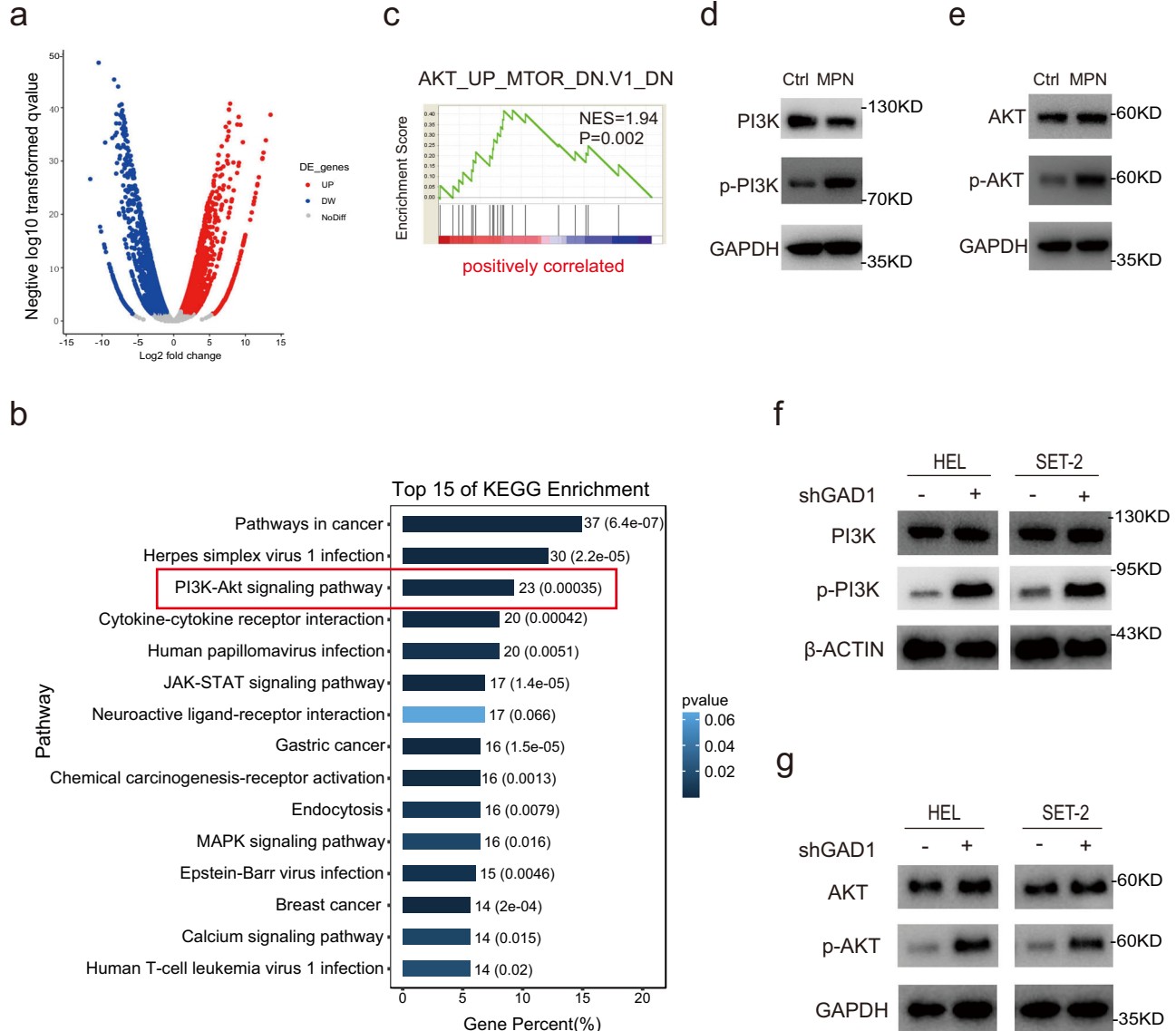

**Fig. 5 | Gadd45g deficiency induces activation of PI3K-AKT pathway. a** RNA-seq analysis were performed on c-kit$^+$ BM cells from *Gadd45g*$^{+/-}$ mice with MPN and those from Ctrl mice. A total of 7098 (3665 upregulated and 3433 downregulated) were differentially expressed in *Gadd45g*$^{+/-}$ mice with MPN. Volcano plot of normalized gene expression in moribund mice with MPN was shown. **b** KEGG enrichment analysis of the RNA-seq data indicating the activation of PI3K-AKT signaling pathway in *Gadd45g*-deficient cells from mice with MPN. **c** GSEA plot showing positive enrichment of AKT signaling in *Gadd45g*-deficient cells from moribund mice with MPN. The normalized enrichment score (NES) and *P* value were shown.

**d** Western blot analysis of p-PI3K and total PI3K protein levels in c-kit$^+$ BM cells from *Gadd45g*$^{+/-}$ mice with MPN and Ctrl mice. **e** Western blot analysis of pAKT-Ser473 and total AKT protein levels in c-kit$^+$ BM cells from *Gadd45g*$^{+/-}$ mice with MPN and Ctrl mice. **f** Western blot analysis of p-PI3K and total PI3K protein expression in HEL and SET-2 cells with or without *GADD45g* knockdown. **g** Western blot analysis of pAKT-Ser473 and total AKT protein expression in HEL and SET-2 cells with or without *GADD45g* knockdown. For (**b**, **c**): Two-tailed Student's *t* test, no adjustment was made for multiple comparisons. For (**d**–**g**): At least three independent experiments were performed.

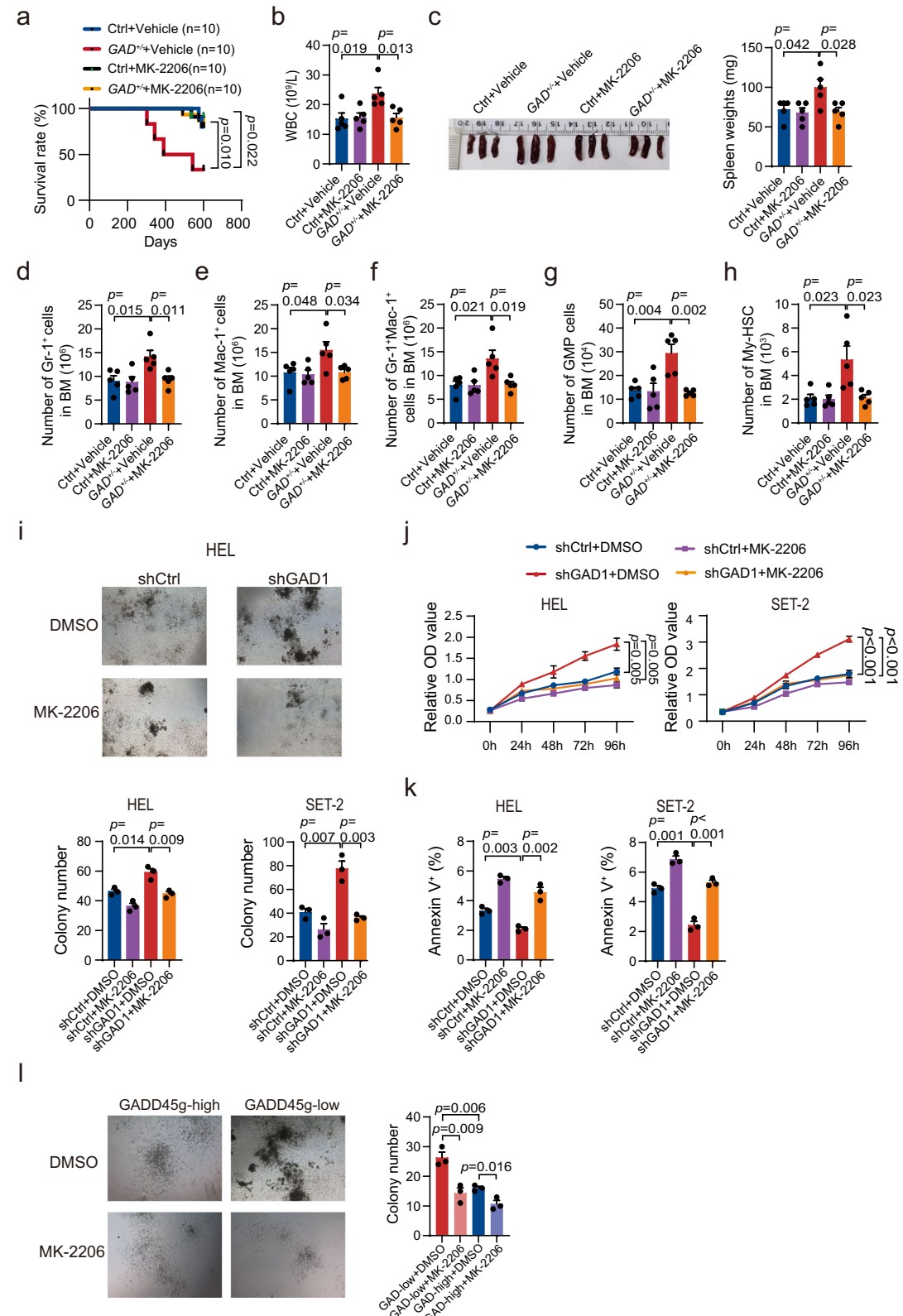

## Human samples

Bone marrow specimens were collected from healthy adult donors and patients with newly diagnosed MPNs at the Institute of Hematology and Blood Diseases Hospital, Chinese Academy of Medical Sciences. The patient cohort consisted of 21 males and 14 females with a median age of 53 years (range: 14-72 years). 16 patients were diagnosed with ET, 19 patients were diagnosed with PV. Human cord blood

(CB) was obtained from healthy postpartum women. All primary samples from human subjects were obtained after informed consent.

## Drugs and chemicals

AKT inhibitor MK-2206 2HCl, PAK1 inhibitor IPA-3, RAC inhibitor EHT 1864 2HCl, JAK2 inhibitor ruxolitinib, and HDAC1/2 inhibitor Romidepsin were all purchased from SelleckChem (Houston, Texas, USA). 5-

**Fig. 6 | Inhibition of PI3K-AKT pathway eliminates the tumor-promoting effects of *Gadd45g* deficiency in MPN. a–h** Two-month-old *Gadd45g*[+/–] and Ctrl mice were orally administered with vehicle or MK-2206 at 100 mg/kg 3 times a week for 2 months, followed by once a week for another 2 months (No overt toxicity was observed at these doses and schedule). Kaplan-Meier survival curves of mice in each group (*n* = 10 mice per group, log-rank test) (**a**). Counts of WBC in the PB of mice in each group (*n* = 5 mice per group) (**b**). Representative images of spleens (left) and spleen weights (right) of mice in each group (*n* = 5 mice per group) (**c**). Absolute number of Gr-1[+] cells (**d**), Mac-1[+] cells (**e**), Gr-1[+]Mac-1[+] cells (**f**), GMP cells (**g**), and My-biased HSCs (**h**) in the BM of mice in each group (*n* = 5 mice per group).

**i** Effects of MK-2206 treatment (10 µM for 14 days) on colony formation of *GADD45g* knocked down HEL and SET-2 cells. **j–k** HEL and SET-2 cells with or without *GADD45g* knockdown were incubated with MK-2206 (10 µM) or DMSO as control for 72 h. Effects of MK-2206 treatment on proliferation (**j**) and apoptosis (**k**) of these cells. **l** Effects of MK-2206 treatment (10 µM for 14 days) on colony formation of primary BM CD34[+] cells from GADD45g[high] and GADD45g[low] patients with MPN (*n* = 3 patients per group). For (**i–k**) Figures shown are representative of three independent experiments with similar results. Data are shown as mean ± SD (*n* = 3 technical replicates). Comparisons were evaluated by two-tailed Student's *t* test, and multiple groups were analyzed with one-way ANOVA.

Aza-2′-Deoxycytidine were purchased from Sigma-Aldrich (St. Louis, Missouri, USA). Doxycycline (Dox, hyclate) were purchased from MedChemexpress (Princeton, New Jersey, US).

### Complete blood counts, morphological and histological analysis

For PB cellularity determination, venous blood collected from the facial vein of living mice was subjected to automated blood cell count (SYSMEX XN-1000 Sysmex's flagship analyzer, Kobe, Hyogo, Japan). For morphological analysis, PB smears were subjected to Wright-Giemsa staining. For histological analysis, femurs and spleens were fixed in 4% paraformaldehyde and embedded in paraffin. Sections (5 µm) were stained with hematoxylin and eosin (H&E).

### Flow cytometry analysis and cell sorting

The BM cell suspensions were flushed from femurs and tibiae. Spleen cells were pestled by the plug of a 10 mL syringe and then filtered. Cells were incubated with panels of fluorochrome-conjugated antibodies. Flow cytometry was performed on a FACS Canto or a FACS LSR II (BD Biosciences, Franklin Lake, NJ, USA). Data analysis was carried out using Diva (BD Biosciences) or FlowJo software (Tree Star, Inc., Ashland, OR). In order to isolate LT-HSCs, My-biased HSCs, c-kit[+] or human CD45[+] (hCD45[+]) cells, BMMNCs were collected and enriched for c-Kit[+] cells or hCD45[+] cells by magnetic bead selection (CD117 MicroBeads, mouse, 130097146; CD45 MicroBeads, human, 130-045-801, Miltenyi Biotec, Bergisch Gladbach, Germany). The enriched cells were then stained with conjugated antibodies and sorted on a FACS Aria III instrument (BD Biosciences). More details on cell sorting antibodies and gating strategies are provided in Supplementary Fig. 12 and Supplementary Table 2, 3. Validation statements and dilutions of all antibodies used for flow cytometry were described on the manufacturer's website with relevant citations.

### Homing assay

Two thousand freshly sorted LT-HSCs or My-biased HSCs from *Gadd45g*[+/–], *Gadd45g*[–/–] or control mice were injected intravenously into lethally irradiated recipients. Donor-derived cells were detected by FACS at 18 h after transplantation.

### Competitive transplantation assays

For competitive LT-HSCs or My-biased HSCs transplantation, five hundred freshly sorted LT-HSCs or My-biased HSCs of CD45.2 mice along with $2 \times 10^5$ wild-type competitor cells (CD45.1) were transplanted into lethally irradiated wild-type recipients (CD45.1). Chimerism and lineage contribution in PB from recipient mice were analyzed by FACS every 4 weeks for 4 times. For secondary transplantation, five hundred CD45.2[+] LT-HSCs or My-biased HSCs isolated from primary recipients together with $2 \times 10^5$ competitor cells were transplanted into the secondary recipient mice. Chimerism and lineage contribution in BM from recipient mice were evaluated by FACS at 4 months after primary and secondary transplantation.

### BrdU incorporation assay

Mice were injected intraperitoneally with 50 mg/kg BrdU at 16 h before BM harvest. BrdU incorporation was measured using an APC BrdU Flow Kit (BD Biosciences).

### Constructs, retroviral or lentiviral transduction, and transfection

For human *GADD45g* and *RAC2* knockdown, the shRNA sequences of them were designed using online RNAi design software. The sequences of the shRNA oligos were listed in Supplementary Table 4. The *GADD45g* shRNA and scramble shRNA were cloned into pLKO.1-GFP vector. The *RAC2* shRNA and its scramble shRNA were cloned into pLKO.1-Puro vector. For human and murine *GADD45g* overexpression, the cDNAs for them were cloned into the Plvx-tight-GFP-Puro Tet-on response vectors. For Dox-inducible human *GADD45g* knockdown, the human *GADD45g* shRNA was cloned into the pLKO-Puro Tet-on response vector. For human *JAK2* and *JAK2V617F* overexpression, lentiviral expression plasmids for *JAK2* (CMV-HsJAK2-IRES2-EGFP) and *JAKV617F* (CMV-HsJAK2VF-IRES2-EGFP) were generated by Azenta Life Sciences (Burlington, Massachusetts, USA).

293T cells were transiently transfected with lentiviral vector and packaging plasmids including psPAX2 and pMD2.G, and culture supernatants containing lentiviruses were collected 48 and 72 h after transfection. Cells were transduced with the recombinant lentiviruses by spin infection. shRNA expressing cells were sorted by FACS or selected with puromycin (2 µg/mL). As for the Dox-inducible lentiviral system, puromycin was added to the transfected cells for selection, and GADD45g expression was induced by Dox (2 µg/mL).

### Cell cycle and apoptosis analysis

Cell cycle was detected by PI staining. Cells were fixed with 70% ethanol at 4 °C for 30 min. Next, cells were incubated with RNase A (0.1 mg/mL) at 37 °C for 30 min. Finally, they were stained with PI (100 µg/mL) (Solarbio, Beijing, China) in the dark at 4 °C for 30 min. The percentages of stained cells in each phase were measured by FACS and analyzed by FlowJo software. Cell apoptosis was measured using an Annexin V Apoptosis detection kit, according to the manufacturer's directions (BD Biosciences).

### MPN xenograft model

For xenografting of *JAK2V617F*-mutated human cells, HEL92.1.7-Luc cells were transfected with Dox-inducible *GADD45g*-specific shRNA. $1.5 \times 10^6$ viable cells were then injected into irradiated (1.8 Gy) NSG mice via lateral tail vein injection. The mice were monitored for tumor burden for 10 days before being randomized into two groups and treated with either vehicle or ruxolitinib (90 mg/kg daily, by oral gavage). After 2 weeks of treatment, each of the ruxolitinib and vehicle treatment groups was subdivided into Dox and non-Dox treatment groups. 2 mg/mL of Dox was added to drinking water for mice and maintained through their lifetime to induce *GADD45g* knockdown. Disease status was imaged using Caliper IVIS Lumina II (Caliper Life Sciences, Hopkinton, Massachusetts, USA).

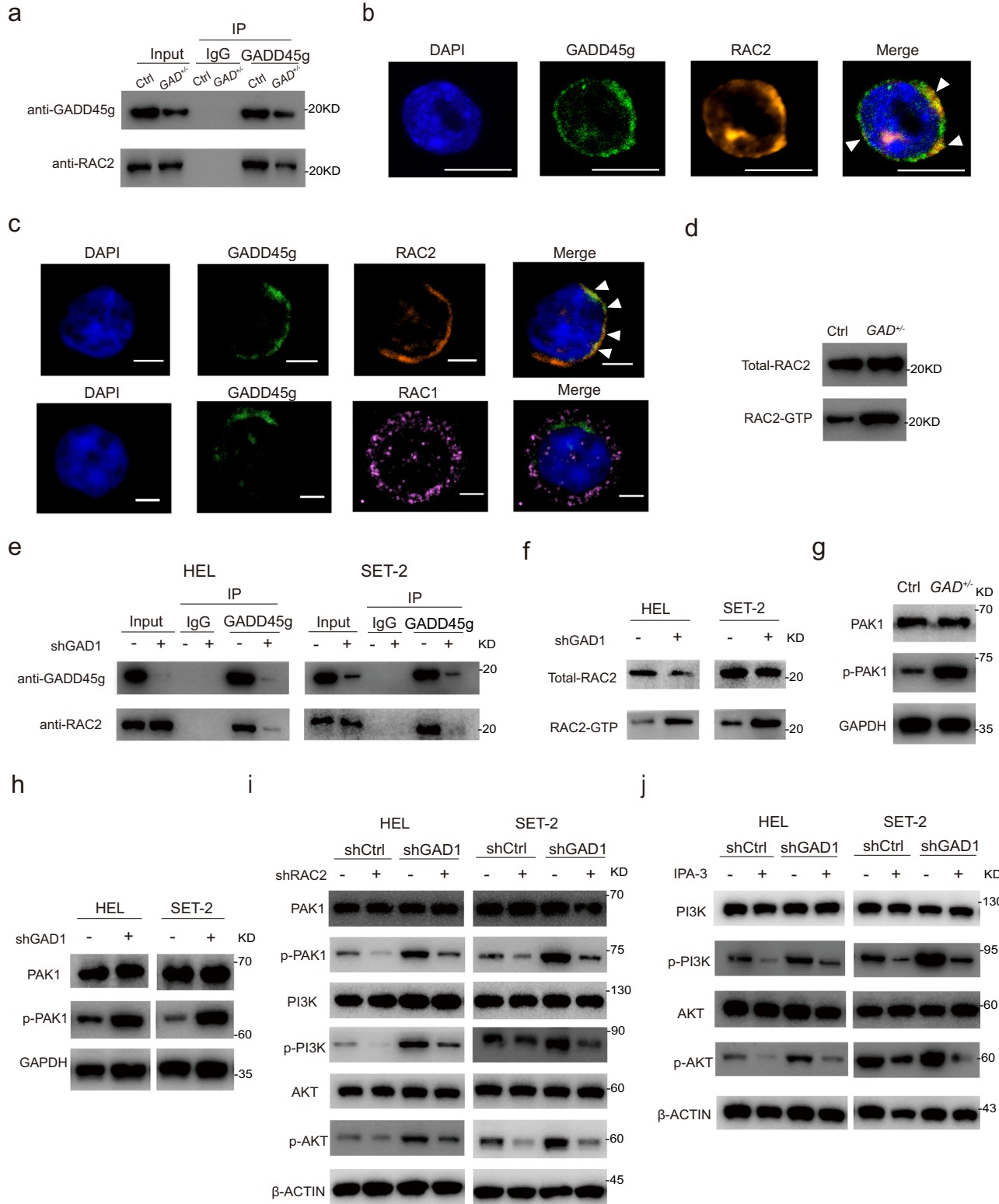

## Cell proliferation assessment

Cell proliferation was estimated by the CCK-8 assay after 72 h of lentivirus transduction or drug treatment. Cells were plated on a 96-well culture dish at a density of $2 \times 10^3$ cells/well. Next, the cells were incubated with CCK-8 solution (Bimake, Houston, Texas, USA) at 37 °C for 4 h. Finally, the optical density (OD) was measured at 450 nm using a standard enzyme instrument.

## Colony formation assays

HEL and SET-2 cells were plated in MethoCult *H4100* (Stemcell, Vancouver, BC, Canada), and colonies were counted on day 14 post plating. Human CD34+ cells were plated in MethoCult *H4434* (Stemcell) in the presence of human FLT3 (Sigma-Aldrich), TPO (Sigma-Aldrich), and SCF (Sigma-Aldrich) and colony numbers were counted after 10-14 days.

**Fig. 7 | RAC2-PAK1 pathway mediates the *Gadd45g* insufficiency-induced activation of PI3K-AKT. a** c-kit[+] BM cells from diseased *Gadd45g*[+/−] and Ctrl mice were lysed, precipitated with anti-GADD45g antibody, and detected by Western blot with anti-RAC2 and -GADD45g antibodies. **b** Representative immunofluorescence micrographs showing colocalization of GADD45g with RAC2 in c-kit[+] BM cells of Ctrl mice. Panels represent nucleus (blue), GADD45g (green), RAC2 (yellow), and merged images, respectively. Arrows in merged image indicate colocalization of GADD45g with RAC2. Bar represents 10 µm. **c** Representative immunofluorescence micrographs showing cellular distribution of GADD45g, RAC2 and RAC1 in cord blood CD34[+] cells from healthy human donors. Panels represent nucleus (blue), GADD45g (green), RAC2 (orange), RAC1 (pink), and merged images, respectively. Arrows in merged image indicate colocalization of GADD45g with RAC2. Bar represents 5 µm. **d** Western blot analysis of RAC2-GTP and total RAC2 protein levels in c-kit[+] BM cells from diseased *Gadd45g*[+/−] and Ctrl mice. **e** HEL and SET-2 cells transfected with GADD45g-specific shRNA or shCtrl were lysed, precipitated with

anti-GADD45g antibody, and detected by Western blot with anti-RAC2 and -GADD45g antibodies. **f** Western blot analysis of RAC2-GTP and total RAC2 protein levels in HEL and SET-2 cells transfected with *GADD45g*-specific shRNA or shCtrl. **g** Western blot analysis of p-PAK1 and total PAK1 protein levels in c-kit[+] BM cells from diseased *Gadd45g*[+/−] and Ctrl mice. **h** Western blot analysis of p-PAK1 and total PAK1 protein levels in HEL and SET-2 cells transfected with *GADD45g*-specific shRNA or shCtrl. **i** HEL and SET-2 cells were transfected with *GADD45g*-specific shRNA or shCtrl for 48 h, followed by transfection with RAC2-specific shRNA (shRAC2) or scrambled control (Scr) for another 48 h. The protein levels of p-PAK1, total PAK1, p-PI3K, total PI3K, pAKT-Ser473 and total AKT were examined by Western blot. **j** HEL and SET-2 cells were transfected with *GADD45g*-specific shRNA or shCtrl for 48 h, followed by treatment with vehicle or IPA-3 (10 µM for 48 h). The protein levels of p-PI3K, total PI3K, pAKT-Ser473 and total AKT were examined by Western blot. For (**a**–**j**): At least three independent experiments with similar results were performed.

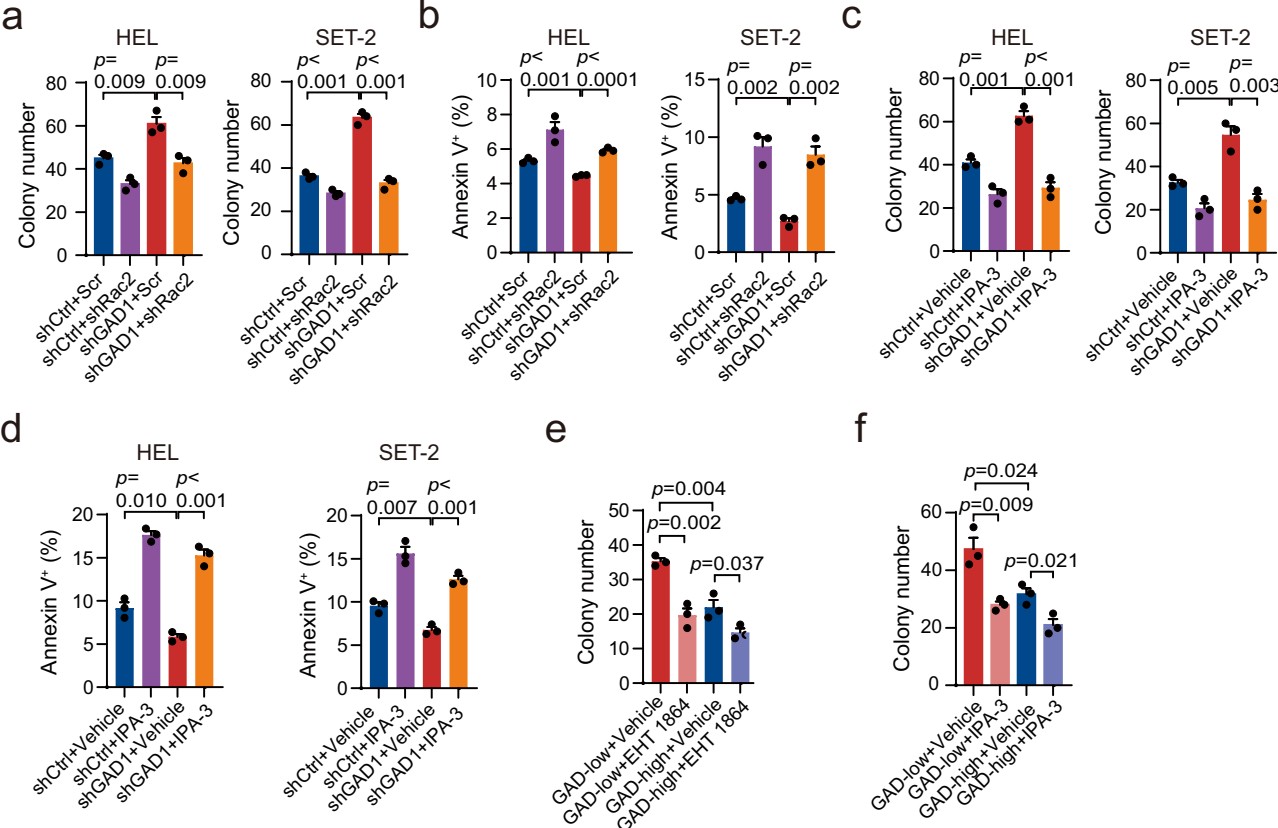

**Fig. 8 | Inhibition of RAC2 or PAK1 reverses the tumor-promoting effects of *GADD45g* deficiency in human MPN cells. a**, **b** HEL and SET-2 cells were transfected with *GADD45g*-specific shRNA or shCtrl for 48 h, followed by transfection with RAC2-specific shRNA (shRAC2) or scrambled control (Scr) for another 48 h. Effects of RAC2 knockdown on colony formation (**a**) and apoptosis (**b**) of these cells. (**c**) HEL and SET-2 cells were transfected with *GADD45g*-specific shRNA or shCtrl for 48 h. Cells were then plated in methylcellulose containing IPA-3 (10 µM). Colonies were counted and analyzed on day 14. **d** HEL and SET-2 cells were transfected with *GADD45g*-specific shRNA or shCtrl for 48 h, followed by treatment with

vehicle or IPA-3 (10 µM) for 48 h, and the percentage of apoptosis cells was examined. **e** Effects of EHT 1864 treatment (5 µM for 10–14 days) on colony formation of primary BM CD34[+] cells from GADD45g[high] and GADD45g[low] patients with MPN (*n* = 3 per group). **f** Effects of IPA-3 treatment (10 µM for 10–14 days) on colony formation of primary BM CD34[+] cells from GADD45g[high] and GADD45g[low] patients with MPN (*n* = 3 per group). For (**a**–**d**): Figures shown are representative of three independent experiments with similar results. Data are shown as mean ± SD (*n* = 3 technical replicates). Comparisons were evaluated by two-tailed Student's *t* test, and multiple groups were analyzed with one-way ANOVA.

## Methylation-specific PCR (MS-PCR) and chromatin immunoprecipitation Quantitative real-time PCR (ChIP-qPCR)

Total genomic DNA from cell lines was extracted using TIANamp Genomic DNA kit (DP304, TIANGEN, Beijing, China). Genomic DNA was treated with sodium bisulfite to convert unmethylated cytosine in CpG sites to uracil using the CpGenome Thrbo Bisulfite Modification Kit (S7847, Sigma-Aldrich). Primers specific for the unmethylated and

methylated promoter sequences used were listed in Supplementary Table 6. MS-PCR assays were performed using Methylation-specific PCR (MSP) Kit (EM101, TIANGEN).

ChIP assays were performed using the Magna ChIP™ A/G Chromatin Immunoprecipitation Kit (17-10085, Sigma-Aldrich) according to the manufacturer's instruction. Anti-acetyl-Histone H3 (Lys9) (9649, Cell Signaling Technology, 10 µl antibody for 10 µg of chromatin), anti-

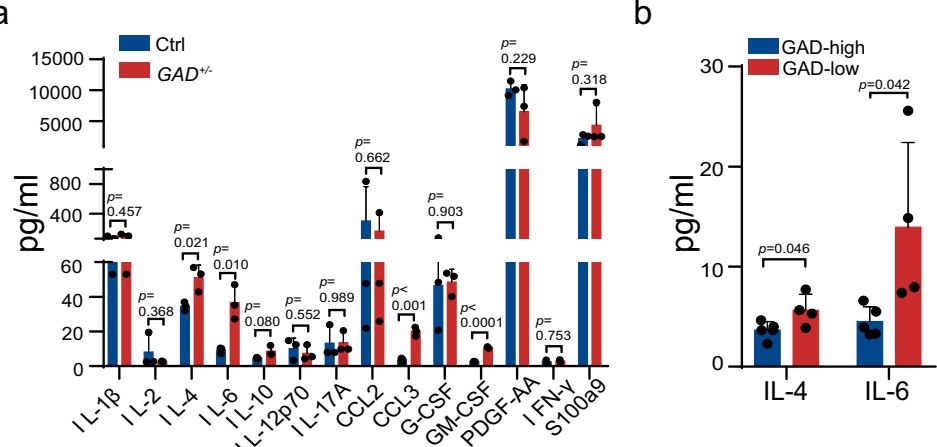

**Fig. 9 | Aberrant production of inflammatory cytokines caused by *Gadd45g* deficiency. a** Serum concentrations of IL-1β, IL-2, IL-4, IL-6, IL-10, IL-12 p70, IL-17A, CCL2, CCL3, G-CSF, GM-CSF, PDGF-AA, IFN-γ and S100a9 in the *Gadd45g^+/−* mice with MPN and their age-matched controls ($n$ = 3 mice per group). **b** Comparisons of serum concentrations of IL-4 and IL-6 between GADD45g^high ($n$ = 5) and GADD45g^low ($n$ = 4) patients with MPN. Data are shown as means ± SD, two-tailed Student's $t$ test.

acetyl-Histone H4 (acetyl K8) (ab45166, Abcam, 2 μg for 25 μg of chromatin), or normal rabbit IgG antibodies (ab172730, Abcam, 1 μg for 25 μg of chromatin) was used for immunoprecipitation. The DNA enriched by ChIP was purified and then quantified by qPCR. The sequences of the primers were listed in Supplementary Table 6.

**Immunoprecipitation, co-immunoprecipitation and mass spectrometry**

Immunoprecipitation (IP) and co-immunoprecipitation (Co-IP) were performed using Pierce Classic Magnetic IP/Co-IP Kit (Thermo Fisher Scientific, Grand Island, NY, USA) according to the manufacturer's directions. Briefly, whole cell protein was extracted using cell lysis buffer for IP (Beyotime, Shanghai, China) supplemented with PMSF (Beyotime) and immunoprecipitated with GADD45g antibody (sc393261, Santa Cruz Biotechnology, Santa Cruz, CA, USA, 1 μg per 200 μg of total protein). For Mass spectrometry assay, the elution products were separated by SDS-PAGE and stained with Pierce™ Silver Stain for Mass Spectrometry Kit (Thermo Fisher). The bands were cut from the *silver*-stained gel and analyzed via liquid chromatography tandem mass spectrometry (LC-MS/MS) in Shanghai Applied Protein Technology (Shanghai, China). For Western blot analysis, the antibodies used are as following: anti-GADD45g (sc393261, Santa Cruz, 1:500 dilution), anti-RAC2 (ab191527, Abcam, Cambridge, MA, USA, 1:1000 dilution) and anti-RAC1 (ab155938, Abcam, 1:1000 dilution).

**Western blot analysis**

Cells were lysed in RIPA (Beyotime) supplemented with PMSF (Beyotime), protease and phosphatase inhibitor cocktail (Beyotime). After lysis on ice for 30 min, the samples were centrifuged, and supernatant were collected. Proteins were detected with primary antibodies against GADD45g (sc393261, Santa Cruz, 1:500 dilution), AKT (9272 S, Cell Signaling Technology, Danvers, MA, USA, 1:1000 dilution), AKT phospho-Ser473 (4060, Cell Signaling Technology, 1:1500 dilution), PI3K (ab154598, Abcam, 1:1000 dilution), PI3K phosphor-Y607 (ab182651, Abcam, 1:500 dilution), PAK1 (2602, Cell Signaling Technology, 1:1000 dilution), PAK1 phospho-Ser144 (2606, Cell Signaling Technology, 1:1000 dilution), RAC2 (ab191527, Abcam1:1000 dilution), GAPDH (2188, Cell Signaling Technology, 1:2000 dilution), JAK2 (3230, Cell Signaling Technology, 1:1000 dilution), phospho-JAK2 (Tyr1007/1008) (3776, Cell Signaling Technology, 1:1000 dilution), β-Actin (ACTBD11B7, Santa Cruz, 1:2000 dilution).

**RAC2 activation assay**

RAC2 activation assay was performed according to the manufacturer's protocol (RAC2 activation assay kit, Millipore, Boston, MA, USA). Briefly, guanosine triphosphate-bound active RAC2 was pulled down with p21-activated kinase-p21-binding domain beads (ab211177, Abcam) for 1 h at 4 °C and subsequently subjected to immunoblot analysis. Immunoblot of whole-cell lysates were used to assess for total RAC2.

**Quantitative real-time PCR**

Total RNA was extracted from isolated cell populations using the TRIzol extraction reagent (Thermo Fisher). Then, the RNA was reverse transcribed to cDNA using M-MLV Reverse Transcript reagent (Thermo Fisher). Quantitative real-time PCR (qRT-PCR) was performed with SYBR Green PCR kit (TaKaRa Bio Inc, Otsu, Shiga, Japan) and analyzed in an ABI 7500 Sequence Detection System. The primers for qRT-PCR are listed in Supplementary Table 5. GAPDH was used as the reference gene for normalization.

**Immunofluorescence**

Cytospins were permeabilized, and fixed in 4% paraformaldehyde/phosphate-buffered saline. GADD45g (sc393261, Santa Cruz), RAC2 antibody (abx001053, Abbexa, Cambridge, UK) and RAC1 antibody (ab97732, Abcam) were used at 1:100 and incubated overnight at 4 °C. After washing, slides were incubated with anti-Rabbit Delight 550 or Anti-mouse Delight 488-conjugated secondary antibody (Thermo Fisher) for 1 h at room temperature. Samples were then incubated with DAPI (1 μg/mL) for 5 min. Images were collected using an UltraVIEW VoX spinning disk confocal system (Perkinelmer).

**Cytokine quantification by Luminex technology**

Serum was obtained after centrifugation at 400 × $g$ for 10 min, snap frozen in liquid nitrogen and stored at −80 °C until further use. The concentration of cytokines in the serum were analyzed using the Mouse Magnetic Luminex Assay (LXSAMSM-22, R&D system, Minneapolis, Minnesota, USA). Briefly, standards and samples were incubated with diluted microparticle cocktail for 2 h at room temperature, followed by incubating with diluted Biotin-Antibody Cocktail for 1 h. After washing, Streptavidin-PE was added to each well and incubated for 1 h at room temperature. Wash steps were performed using the wash buffer. All other steps were performed according to the manufacturer's instructions in LabEx of UNIV (Shanghai, China). Samples

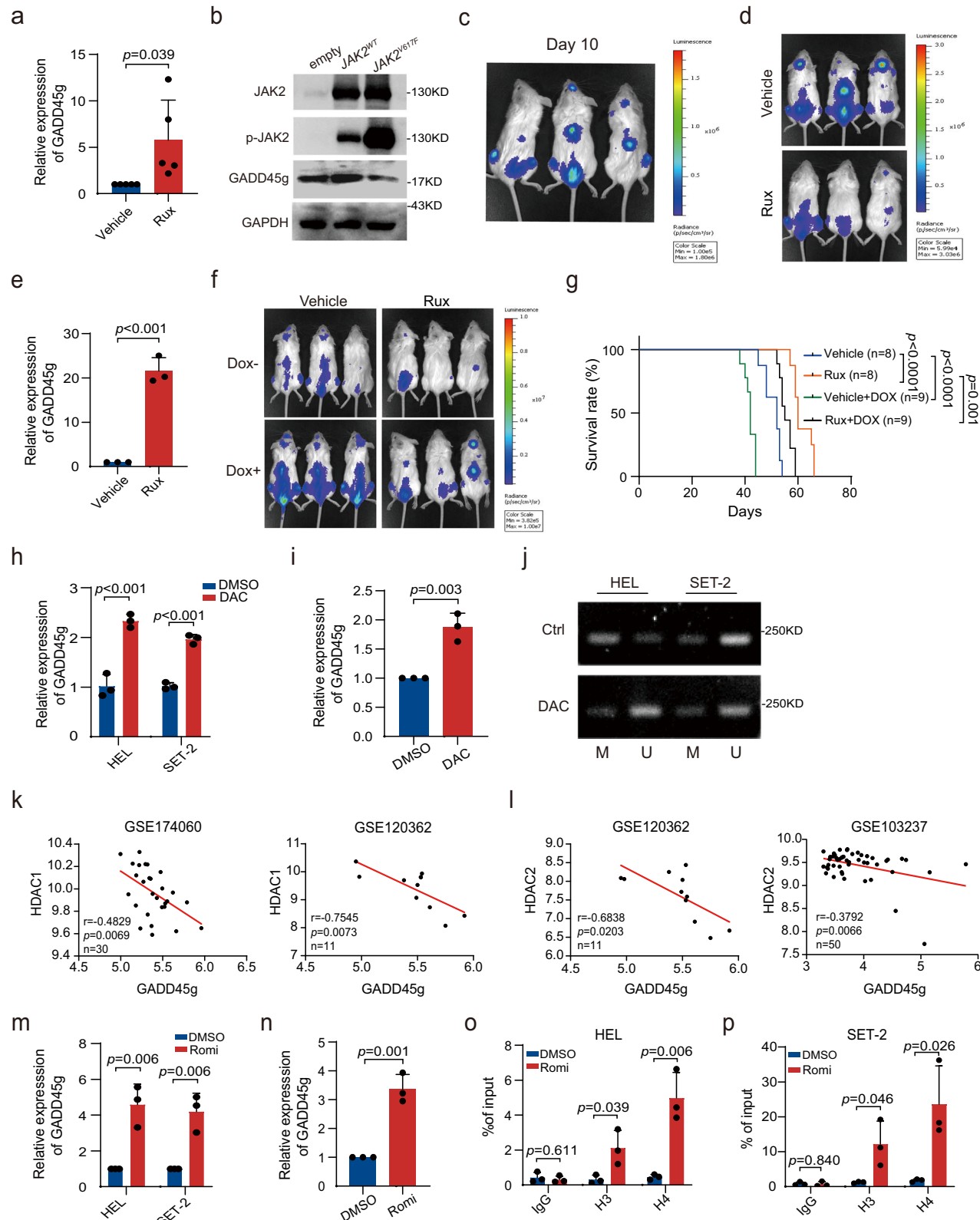

were then analyzed on a Bio-Plex MAGPIX (Luminex/Bio-Rad) following the manufacturer's guidelines.

## RNA-seq and analysis
The c-kit⁺ BM cells were sorted using a BD FACS Aria III instrument (BD Biosciences). Total RNA was extracted using TRIzol reagent (Thermo Fisher) and RNA concentrations were measured by NanoDrop 2000 spectrophotometer (Thermo Fisher). The RNA sequencing library was prepared and sequenced on Illumina HiSeq 4000 platform (Illumina, San Diego, CA, USA). Differentially expressed genes (DEGs) with log2 (fold change) > 1 and FDR < 0.05 were considered significant. KEGG enrichment analyses were performed using DAVID.

**Fig. 10 | The expression of *GADD45g* is repressed by *JAK2V617F* mutation and histone deacetylation, and *GADD45g* reduction plays a tumor-promoting role in *JAK2V617F* MPN. a** qRT-PCR analysis of *GADD45g* expression in BMMNCs from patients with PV treated with 50 nM ruxolitinib (Rux) or vehicle for 24 h (*n* = 5). **b** Western blot analysis of JAK2, p-JAK2 and GADD45g protein levels in 293T cells transduced with lentiviruses expressing *JAK2V617F*, wild-type *JAK2* or empty vector constructs. Blots shown are representative of three independent experiments. **c** HEL92.1.7-Luc cells were transfected with Dox-inducible *GADD45g*-specific shRNA and engrafted into irradiated NSG mice. Bioluminescence imaging of representative mice was taken at day 10 post-transplantation (*n* = 3 mice per group). **d** The NSG mice were treated with either vehicle or Rux for 2 weeks (*n* = 20 mice per group). Bioluminescence imaging of representative mice from each group were taken (*n* = 3 mice per group). **e** qRT-PCR evaluation of *GADD45g* expression in hCD45-positive cells isolated from the BMMNCs of mice in each group (*n* = 3 mice per group). **f, g** Bioluminescence imaging of representative mice from the indicated groups were taken after 1 week of shRNA expression induced by Dox treatment (**f**). Kaplan-Meier survival curves of mice in each group were shown (*n* = 8–9 mice per

group, log-rank test) (**g**). **h–j** MPN cell lines and BMMNCs from MPNs patients were treated with 2 μM Decitabine (DAC) for 96 h. qRT-PCR analysis of *GADD45g* expression in MPN cell lines (**h**) or BMMNCs from MPNs patients (*n* = 3 patients) (**i**). MS-PCR showing the methylated GADD45g alleles in MPN cell lines (M, methylated allele; U, unmethylated allele). Bands shown are representative of three independent experiments (**j**). **k, l** Correlations of the expression between *GADD45g* and *HDAC1* (**k**) or *HDAC2* (**l**) in CD34+ BMMCs/PBMCs from MPNs patients. (**m**) qRT-PCR evaluation of *GADD45g* expression in MPN cell lines treated with 3 nM Romidepsin (Romi) or DMSO for 48 h. **n** qRT-PCR analysis of *GADD45g* expression in BMMNCs from MPNs patients treated with 5 nM Romi or DMSO for 48 h (*n* = 3). **o, p** HEL (**o**) and SET-2 cells (**p**) were treated with 3 nM Romi or DMSO for 48 h, and then subjected to ChIP analysis. The enriched DNA which associated with the promoter region of GADD45g was quantified by qPCR. For (**a, e, h, i, k, p**): Data are shown as means ± SD, two-tailed Student's *t* test. For (**h, m, o, p**): Figures shown are representative of three independent experiments with similar results. *n* = 3 technical replicates.

## Whole exome sequencing

Genomic DNA extracted from c-kit+ BM cells was fragmented to an average size of 180 - 280 bp and subjected to DNA library creation using established Illumina paired-end protocols. The Agilent SureSelect Mouse All ExonV6 Kit (Agilent Technologies, Santa Clara, CA, USA) was used for exome capture according to the manufacturer's instructions. DNA sequencing was performed at Novogene Bioinformatics Technology (Beijing, China), using an Illumina Novaseq 6000 platform (Illumina Inc., San Diego, CA, USA) to generate 150-bp paired-end reads.

## Statistics and reproducibility

All statistical data analysis were performed using GraphPad Prism v.8 (GraphPad Software). No statistical method was used to predetermine sample size. No data were excluded from the analyses. For animal experiments, mice with the same gender and age were divided randomly into experimental groups to reduce variation and enhance power. For some experiments, such as the Western blot assays, the investigators were not blinded to allocation during the experiments and outcome assessment. For animal experiments related to pathologic diagnosis, the investigators were blinded to avoid subjective judgment. Results are displayed as the mean ± SD. Student's *t* test and one-way ANOVA were used for statistical comparisons where appropriate. Survival was analyzed by the Kaplan-Meier method and compared by the log-rank test.

## Reporting summary

Further information on research design is available in the Nature Portfolio Reporting Summary linked to this article.

## Data availability

The RNA-Seq data generated in this study have been deposited in NCBI's GEO repository under accession number GSE229495. The WES data generated in this study have been deposited in NCBI's Sequence Read Archive under accession number PRJNA918529. The mass spectrometry proteomics data generated in this study have been deposited to the ProteomeXchange Consortium via the iProX partner repository[46,47] with the dataset identifier PXD039261 [https://www.iprox.cn//page/project.html?id=IPX0005710000]. The published GEO datasets used in this study are available under accession number GSE53482, GSE174060, GSE120362, and GSE103237. Source data are provided with this paper.

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

## Acknowledgements

This work was supported by grants from CAMS Innovation Fund for Medical Sciences (2021-I2M-1-019, X.M.), National Natural Science Foundation of China (82270122, X.M.; 82070113, X.M.; 81890990, T.C.; 81730006, T.C.).

## Author contributions

X.M., T.C., Z.X. and L.Z. conceived the project and designed the studies. P.Z., N.Y., Y.D., W.Z. and N.W. performed experiments, interpreted results, and wrote the manuscript. Y.D. and W.Z. constructed the xenograft model. Y.X. and W.H. performed animal experiments and analyzed data. Q.R., T.Q. and R.F. performed cell experiments, immunohistochemistry, Western blot, and analyzed data. All authors approved the final version of the manuscript.

## Competing interests

The authors declare no competing interests.
