## [Peer Review File · Nature Communications]

REVIEWER COMMENTS

Reviewer #1 (Remarks to the Author):

This is an interesting report of GADD45g deficiency leading to myeloproliferative neoplasm/like disease. Overall the work is of high technical quality and general interest.

The paper is data dense and my overall major comment is that it is difficult to read/follow. Some reorganisation and prioritisation of figures would be beneficial in this regard. Interestingly, the phenotype appears identical between the heterozygous and homozygous loss.

Specific comments

1. GADD45g is not mutated in AML(SF1A), however there are some data in figure 3A-E which show downregulation of expression and even protein in MPN. These data should be brought together and provided in the main figure 1. Ideally this should be combined with the murine data showing a MPN-like phenotype (figure 2). Figure 3A does not appear convincing, however the other panels are more convincing.

2. Why are the MPN-like mice anemic, despite expansion of red cell precursors? The fibrosis data shown in SF6 is not convincing, appears to be only from 1 mouse and there is no reticulin stain provided. Is MF a feature of this model?

3. Gain of function of HSC in transplantation setting. The methods are clear, however the text accompanying the current Figure 1 is difficult to follow. It appears that all the transplants (Fig1) are performed only with competitive LTHSC vs. helper transplants. All of the analysis of primary mice has been relegated to supplementary data.

e.g. "The hematological parameters of Gadd45g-deleted and Ctrl mice were examined every two months after birth. Both Gadd45g+/- and Gadd45g-/- mice exhibited normal hematopoiesis until 4 months of age when the self-renewal capacities of their long-term HSCs (LT-HSC) were significantly increased (Fig. 1A, Supplementary Fig. 3A)." - however data from 2 month and 4 month mice are not shown, and SF4 shows only data from 6 month non-transplanted mice.

Minor comment relating to figure 1 - why are the axes not linear in figure 1F and 1G?

4. Supplementary Figure 4H and 4I should be shown in primary data

5. MPN-like disease is transplantable, and rescued by re-expression of GADD45g. These data from SF7 and SF8 are important and should be shown in primary data with figure 2.

6. The immediate question from a transplantable myeloid malignancy is whether additional mutations are present. This is addressed in Supplementary Figure 14, however should be referenced and presented together with Figure 2 as well.

7. There is activation of PI3K and AKT and treatment with MK2206 was able to prolong survival and reduce disease parameters. The authors have referenced this. Similarly, the role of PAK and RAC2 interaction is well established in HSC and should be noted:
<https://pubmed.ncbi.nlm.nih.gov/23335370/>

8. the MPN appears to be driven by cytokine inflammation, especially IL6. Can the disease parameters be reversed by agents that block cytokine signaling, either anti IL6 or JAK inhibitors?

Reviewer #2 (Remarks to the Author):

In this manuscript authors have examined the role of GADD45g in driving MPN. They have generated a mouse model lacking Gadd45g and show that its loss causes a MPN like phenotype but not AML in mice. Based on these studies they claim that loss of GADD45g drives MPN in humans. They manipulate 2 MPN cell lines to make this point.

Critique:

Authors have previously shown that GADD45g is also downregulated in AML (Blood, 2021). In those studies, using AML cell lines they showed a role for p38 MAPK in antileukemic activities of GADD45g. They also showed a role for FLT3ITD and MLLAF9 in driving this process. Thus, it appears that GADD45g is generally repressed in myeloid malignancies and in fact similar findings have been shown in solid tumors. The fact that 90 to 95% of MPNs are driven by defined mutations, PI should experimentally explore the relationship between these mutations and GADD45g expression in MPNs. How do these mutations impact GADD45g expression in MPNs and are the mechanisms different between different myeloid malignancies (i.e. AML vs. MPN) otherwise the study lacks novelty. Further, what is the level of suppression of GADD45g in primary MPNs driven by different mutations – JAK, MPL, TET2, DNMT3A etc. Is there any variation in the expression between different mutant bearing MPNs or their combinations? These data should be included. Furthermore, PDX studies in NSGS mice should be performed to determine the extent to which restoration of GADD45g restores MPN in the presence and/or absence of existing therapies such as JAK2 inhibitors and in MPNs with different mutations. These types of studies would help in better defining the role of GADD45g in MPN development. As presented, the link between MPN and GADD45g is tenuous at best.

Other comments:

Bars in Figure 1 look overlapping. The differences in engraftment appear modest. Engraftment analysis at 32 weeks should also be included. Have the authors looked at the homing of the knockout cells? This could help explain the modest difference in engraftment.

As pointed out above, the mechanism of GADD45g repression in MPNs and is this mutation dependent should be explored further.

In PDX models of MPN does combination of AKT and JAK2 or RAC inhibitors correlate with amount of GADD45g repression in terms of disease burden and stem cell function in vivo.

A recent study showed activation of Rac in Dnmt3a mutant cells and repression of AML/MPN like disease by Rac inhibitor in vivo; do Dnmt3a mutations repress GADD45g? if so, how?

Does GADD45g bind to RAC2 but not RAC1 in primary patient derived MPNs? How is this specificity achieved? How does binding of the two or lack of result in increased Rac activation?

Reviewer #3 (Remarks to the Author):

In the current study the authors demonstrate that Gadd45g insufficiency in the murine hematopoietic system leads to significantly enhanced growth and self renewal capacity of myeloid-biased HSCs, and the development of phenotypes resembling MPNs. They also provide data to indicate that In humans, GADD45g is expressed at lower levels in patients with MPNs, and its down regulation in MPN cells plays tumor-promoting roles. Mechanistically, the pathogenic role of GADD45g insufficiency is shown to be mediated via a cascade of activations of RAC2, PAK1 and PI3K-AKT signaling pathways.

Though as a whole the study appears to be well conducted there are some major issues that impacts on its significance:

1. Most of the results are based on Gadd45 KO in the myeloid compartment which does not reflect the physiological levels of Gadd45g in HSC.
2. In a previous study (Blood Vol 138 #6 464-79, 2021) the authors show that GADD45g acts as a novel tumor suppressor in AML , where it exerts its effects via inhibition of E2F1 via the p38 MAPK–dependent signaling pathway and that silencing of GADD45g in AML is associated with epigenetic regulation and relevant leukemic oncogenes. The authors provide no data to explain this apparent dichotomy in Gadd45g signaling in HSC versus AML. This is puzzling and needs to be addressed.

RESPONSE TO REVIEWERS' COMMENTS

Responses to Reviewer #1

Specific comments

1. “GADD45g is not mutated in AML(SF1A), however there are some data in figure 3A-E which show downregulation of expression and even protein in MPN. These data should be brought together and provided in the main figure 1. Ideally this should be combined with the murine data showing a MPN-like phenotype (figure 2). Figure 3A does not appear convincing, however the other panels are more convincing.”

Response:

We appreciate the reviewer's suggestion. However, the Figure Guidelines of *Nature Communications* require that each figure should be no larger than a single A4 page. This study includes numerous works, and each of the original Figure 1 and 2 was already or nearly A4 size, so we cannot combine the mouse and human data into one figure as suggested by the reviewer. In addition, according to our study design, we were initially interested in what disease could be induced in the mouse by altering *Gadd45g*, which may imply the importance of *Gadd45g* in the pathogenesis of this disease. We believe that these data are more important and more compelling than the study of which human disease has a lower expression of *GADD45g*. Therefore, we presented the mouse data in original Figure 1-2, before validating the association of GADD45g with human MPNs. In addition, the mutation analysis of GADD45g in AML, MPN and MDS (SF1A) is intended to ensure the proper way to establish the mouse model that can accurately mimic the alteration of *GADD45g* in patients with myeloid malignancies, which we believe should be presented prior to the mouse knockout data. Based on the above, we have retained the order and most of the content of the figures mentioned by the reviewer in the revised manuscript.

As suggested by the reviewer, the unconvincing Figure 3A, in which the sample size is relatively small and the heterogeneity of the data is large, has been removed. However, we find that no eligible dataset is available to replace it. Therefore, in the revised manuscript, we have moved the results of examination of *GADD45g* expression

in patients with MPNs at our hospital to the front, and the remaining data from public dataset to the back. The results are presented as follows: “To assess whether the findings in mouse are true for human, we first examined the expression levels of GADD45g in patients with MPNs. The results revealed that its expression was significantly lower in primary CD34⁺ and bone marrow mononuclear cells (BMMNCs) from patients with ET and PV, compared to those from healthy volunteers, at both mRNA and protein levels (Fig. 4a-c). In a separate cohort (GSE53482), *GADD45g* was markedly downregulated in CD34⁺ cells from patients with PV and PMF compared to healthy individuals (Fig. 4d).”

2. “Why are the MPN-like mice anemic, despite expansion of red cell precursors? The fibrosis data shown in SF6 is not convincing, appears to be only from 1 mouse and there is no reticulin stain provided. Is MF a feature of this model?”

Response: We appreciate the reviewer’s comment. Although the MPN-like mice exhibited reduced erythropoiesis in the BM, red cell precursors and Ter119⁺ cells were significantly increased in the spleen; however, anemia was still evident. Such phenotypes have often been reported in MPN (PMID:23221337, PMID:28232469, PMID:26834157). A possible explanation for this puzzle is that the increased splenic erythropoiesis in myeloproliferative disorders may be ineffective and may contribute to the anemia (PMID:23221337, PMID: 2757009).

The fibrosis data shown in SF6 were indeed from only 1 mouse, and MF is not a feature of this model. In the revised manuscript, we have removed the description of reticulin staining and the corresponding figure and figure legend.

3. “Gain of function of HSC in transplantation setting. The methods are clear, however the text accompanying the current Figure 1 is difficult to follow. It appears that all the transplants (Fig1) are performed only with competitive LTHSC vs. helper transplants.

Response: We apologize for the confusing display. In the revised manuscript, we have

added the following description of the transplantation setting to the Results section on page 6, paragraph 2: “Serial competitive transplantation assays were performed to assess the function of long-term HSCs (LT-HSC). Briefly, sorted LT-HSCs (CD45.2) were mixed with competitor cells (CD45.1) and then transplanted into CD45.1 recipient mice (Supplementary Fig. 3a). The results revealed that.....”. In addition, we have replaced the original schematic representation of the BM transplantation (Supplementary Fig. 3A) with a more accurate one.

All of the analysis of primary mice has been relegated to supplementary data.

Response: The Figure Guidelines of *Nature Communications* state that the main text should contain no more than 10 figures, and this study includes a lot of work with over 20 figures. Therefore, we need to present some primary mouse data, which we feel is not the most important, in supplementary figures.

e.g. "The hematological parameters of *Gadd45g*-deleted and Ctrl mice were examined every two months after birth. Both *Gadd45g*^{+/-} and *Gadd45g*^{-/-} mice exhibited normal hematopoiesis until 4 months of age when the self-renewal capacities of their long-term HSCs (LT-HSC) were significantly increased (Fig. 1A, Supplementary Fig. 3A)." - however data from 2 month and 4 month mice are not shown, and SF4 shows only data from 6 month non-transplanted mice.

Response: We apologize for the inappropriate description. In the revised manuscript, we changed the sentences mentioned by the reviewer into “The hematological parameters of *Gadd45g*-deleted and Ctrl mice were examined every two months after birth. Both *Gadd45g*^{+/-} and *Gadd45g*^{-/-} mice exhibited normal hematopoiesis at 2 and 4 months of age (data not shown). Serial competitive transplantation assays were performed to assess the function of long-term HSCs (LT-HSC). Briefly, sorted LT-HSCs (CD45.2) were mixed with competitor cells (CD45.1) and then transplanted into CD45.1 recipient mice (Supplementary Fig. 3a). The results revealed that the self-

renewal capacities of LT-HSCs remained unchanged in the 2-month-old *Gadd45g^{+/-}* and *Gadd45g^{-/-}* mice (data not shown), while increased significantly in the 4-month-old mice (Fig. 1a).” Since the manuscript already contained many figures, we have not added the above-mentioned negative results from 2- and 4-month-old mice to the revised manuscript, but have indicated them as “data not shown” and are showing them below for the reviewer.

Figure 1 for Reviewer #1

Figure 1. Hematological parameters of PB and BM of 2- and 4-month-old mice are not significantly altered by *Gadd45g* deletions.

(A) Counts of white blood cell (WBC), platelet, red blood cell (RBC), hemoglobin, and MCV, and percentages of monocytes, neutrophils, and lymphocytes in the PB of 2- and 4-month-old *Gadd45g*$^{+/-}$(*GAD*$^{+/-}$), *Gadd45g*$^{-/-}$ (*GAD*$^{-/-}$) and Ctrl mice (n=5 mice per group).

(B) Total number of BM cells in 2- and 4-month-old *Gadd45g*$^{+/-}$, *Gadd45g*$^{-/-}$ and Ctrl mice (n=5 mice per group).

(C-M) Absolute numbers of Gr-1$^{+}$(C) , Mac-1$^{+}$ (D), Gr-1$^{+}$Mac-1$^{+}$ (E), B220$^{+}$ (F), CD3$^{+}$ (G), Ter119$^{+}$ (H), CMP (I), GMP (J), MEP (K), CLP (L), and HSCs (defined as CD34-CD150+LSK) (M) in the BM of 2- and 4-month-old *Gadd45g*$^{+/-}$, *Gadd45g*$^{-/-}$ and Ctrl mice (n=5 mice per group).

Data are shown as means ± SD. ns, not significant (Student's *t* test).

## Figure 2 for Reviewer #1

Figure 2. Hematological parameters of the spleens from 2- and 4-month-old mice are not significantly altered by *Gadd45g* deletions.

(A) Spleen weights of 2- and 4-month-old *Gadd45g*^{+/-}, *Gadd45g*^{-/-} and Ctrl mice (n=5 mice per group).

(B-L) Absolute numbers of Gr-1⁺ (B), Mac-1⁺ (C), Gr-1⁺Mac-1⁺ (D), B220⁺ (E), CD3⁺ (F), Ter119⁺ (G), CMP (H), GMP (I), MEP (J), CLP (K), and HSCs (L) in the spleens of 2- and 4-month-old *Gadd45g*^{+/-}, *Gadd45g*^{-/-} and Ctrl mice (n=5 mice per group).

Data are shown as means ± SD. ns, not significant (Student's *t* test).

Figure 3 for Reviewer #1

Figure 3. Deletions of *Gadd45g* have no effect on the self-renewal capacities of LT-HSCs from 2-month-old mice.

(A) Percentage of donor chimerism in the PB of lethally irradiated primary recipients transplanted with LT-HSCs (defined as CD34⁻Flk2⁻LSK) freshly isolated from 2-month-old *Gadd45g*^{+/-}, *Gadd45g*^{-/-} and Ctrl mice together with competitor cells. Engraftment was examined monthly after transplantation (n=5 mice per group).

(B) Donor chimerism in the PB of secondary recipients transplanted with CD45.2⁺ LT-HSCs from primary recipient mice together with competitor cells. Engraftment was examined monthly after transplantation (n=5 mice per group).

Data are shown as means \pm SD. ns, not significant (Student's *t* test).

Minor comment relating to figure 1 - why are the axes not linear in figure 1F and 1G?"

Response: We apologize for the carelessness. We have corrected the axes in original Figures 1F and 1G, which have been changed to Fig. 1h and 1i, respectively, in the revised manuscript.

4. "Supplementary Figure 4H and 4I should be shown in primary data"

Response: We appreciate the reviewer's comments. As suggested by the reviewer, in the revised manuscript, the original Supplementary Figures 4H and 4I have been moved to the main Figure 1 and renumbered as Fig. 1b and 1d, respectively.

5. "MPN-like disease is transplantable, and rescued by re-expression of GADD45g. These data from SF7 and SF8 are important and should be shown in primary data with figure 2."

Response: We appreciate the reviewer's comments. According to The Figure Guidelines of Nature Communications, each figure should be no larger than a single A4 page. Since Figure 2 already contained 14 panels and was almost A4 size, we could not fit all of SF 7 and SF 8 into it. Instead, we moved SF 7A and SF 8, which we considered were relatively more important, to Figure 2 and renumbered them as Fig. 2o and 2p, respectively. The original Supplementary Figures 7B-J were still shown in supplementary data and have been renumbered as Supplementary Fig. 7a-i.

6. "The immediate question from a transplantable myeloid malignancy is whether additional mutations are present. This is addressed in Supplementary Figure 14, however should be referenced and presented together with Figure 2 as well."

Response: We agree with the reviewer's comment. However, Figure 2 is already A4

size, we cannot present SF14 together with it. Thus, we added a new figure, numbered as Figure 3, and presented the original Supplementary Figure14E-G, the data regarding additional mutations, as Fig. 3a-c. The original Figure 3 and subsequent figure numbers have been changed accordingly. In addition, the description of the whole-exome sequencing results has been moved from the Results section on page 16, paragraph 2 in the original manuscript to Results section on page 10, paragraph 2 in the revised manuscript.

7. “There is activation of PI3K and AKT and treatment with MK2206 was able to prolong survival and reduce disease parameters. The authors have referenced this. Similarly, the role of PAK and RAC2 interaction is well established in HSC and should be noted: <https://pubmed.ncbi.nlm.nih.gov/23335370/>”

Response: As suggested by the reviewer, the paper <https://pubmed.ncbi.nlm.nih.gov/23335370/> has been added as reference 21 in the revised manuscript. The original reference 21 and subsequent reference numbers have been changed accordingly.

8. “the MPN appears to be driven by cytokine inflammation, especially IL6. Can the disease parameters be reversed by agents that block cytokine signaling, either anti IL6 or JAK inhibitors?”

Response: We appreciate the reviewer’s comments. As suggested by the reviewer, we assessed the effects of IL-6 blockade on the progression of MPN induced by *Gadd45g* insufficiency. 10-month-old *Gadd45g*-deficient mice with MPN-like phenotypes (leukocytosis, thrombocytosis and anemia) and age-matched Ctrl were randomized and intraperitoneally treated with IL-6-neutralizing antibody (R&D Systems, MAB406) or control IgG (R&D Systems, 6-001-A) at 1 mg/kg twice a week for 2 months. The opted dose and schedule have been previously reported and no overt toxicity has been observed (PMID: 25337873, PMID: 25159156). The disease burden was analyzed by

complete blood count and flow cytometry. The results showed that only WBC counts and monocyte proportions were significantly reduced by pharmacological inhibition of IL-6, the expansion of My-biased HSCs or the myeloid differentiation bias were not rescued in the BM of *Gadd45g*-deficient mice with MPN. These data suggest that inhibition of IL-6 only minimally reverses the disease parameters in *Gadd45g*-deficient MPN mice. Due to length limitation, these results are not added to the revised manuscript. They are shown in the figure below.

Figure 4 for Reviewer #1

Figure 4. Pharmacologic inhibition of IL-6 modestly affects disease parameters in *Gadd45g*-deficient MPN mice.

(A) Counts of WBC, platelet, RBC, hemoglobin and MCV, and percentages of monocytes, neutrophils and lymphocytes in the PB of mice in each group (n= 4-5 mice per group).

(B-I) Absolute number of CD41⁺CD61⁺ (B), Gr-1⁺ (C), Mac-1⁺ (D), Gr-1⁺Mac-1⁺ (E), Ter119⁺ (F), GMP (G), MEP (H) cells and My-HSCs (I) in the BM of mice in each group (n= 4-5 mice per group).

Data are shown as means \pm SD. *, $P < 0.05$; **, $P < 0.01$; ***, $P < 0.001$; ns, not significant (Student's *t* test).

Responses to Reviewer #2

Major comments:

1. How do these mutations impact *GADD45g* expression in MPNs and are the mechanisms different between different myeloid malignancies (i.e. AML vs. MPN) otherwise the study lacks novelty. What is the level of suppression of *GADD45g* in primary MPNs driven by different mutations – JAK, MPL, TET2, DNMT3A etc. Is there any variation in the expression between different mutant bearing MPNs or their combinations? These data should be included.

Response: We appreciate the reviewer's comments. Our previous study has shown that the oncogenes FLT3-ITD and MLL-AF9, which present with a high leukemic burden and confer a poor prognosis in patients with AML, as well as epigenetic regulation contributed to the silencing of *GADD45g* in AML (PMID: 33945602). To determine the difference between AML and MPN on the mechanisms regulating *GADD45g* expression, we first investigated whether the mutations critical in initiating and driving the development of MPNs are involved in the downregulation of *GADD45g* in MPNs. *JAK2V617F* mutation is the most recurrent mutation driving all the three major entities of MPNs and its high frequency (95% of patients with PV) and the availability of inhibitors allow us to study its effect on *GADD45g* expression. We collected primary bone marrow specimens from patients with newly diagnosed PV (n=5), isolated bone marrow mononuclear cells (BMMNCs) and treated the cells with JAK2 inhibitor (ruxolitinib, Selleck, S1378, 50 nM) or vehicle for 24 hours. qRT-PCR analysis showed that the expression levels of *GADD45g* were markedly upregulated upon ruxolitinib treatment. We next transduced 293T cells with lentiviruses expressing *JAK2V617F*, wild-type *JAK2*, or empty vector constructs, and observed that only *JAK2V617F* overexpression resulted in a significant decrease in the expression level of *GADD45g*. These data suggest that *GADD45g* expression is repressed by *JAK2V617F* mutation in MPNs.

MPL mutations account for only 1-3% of ET and 5% of PMF (PMID: 16868251),

and *DNMT3A* mutations are present in 5-10% of MPNs (PMID:32823933). Their low frequencies prevent us from obtaining primary cells carrying these mutations. *DNMT3A*, *CALR* and *TET2* are loss-of-function mutations that require overexpression for functional assessment (PMID:32733014, PMID:28028029). However, primary MPN cells cannot tolerate the cytotoxic effects of lentiviral-mediated gene overexpression. Due to the above limitations, we are not able to assess whether *MPL*, *CALR*, *TET2* and *DNMT3A* mutations have any effect on the *GADD45g* expression in primary MPN cells.

Notably, our analysis of the published dataset of GSE54646 reveals that the expression level of *GADD45g* in *JAK2V617F*-mutated MPNs patients with additional *TET2* or del (20q13.11) mutations is significantly lower than in patients with *JAK2V617F* mutation alone (see Figure 1 below), suggesting the possibility that other mutations besides *JAK2V617F* may repress the expression of *GADD45g*. However, the sample size of GSE54646 is relatively small (it is the only public dataset regarding the *GADD45g* expression in MPN patients with multiple mutations), so the analytical result of the dataset is not added to the revised manuscript. It is shown in the figure below (Figure 1).

Figure 1 for Reviewer #2

Figure 1. MPN patients with multiple mutations exhibit lower level of *GADD45g*

expression.

Relative *GADD45g* expression in peripheral blood neutrophils from MPNs patients with *JAK2V617* mutation (n=25) or multiple mutations (*JAK2V617F+TET2/del* (20q13.11), n=7) (GSE54646).

Data are shown as means \pm SD. *, $P < 0.05$ (Student's *t* test)

The inhibitory effect of *JAK2V617F* mutation on *GADD45g* expression, as indicated by our observations, is not consistent with the analytical results from the public datasets presented in our original manuscript, which concluded that the reduced expression of *GADD45g* was irrelevant to somatic mutations (Supplementary Fig.14A-D). A possible explanation is that only mutations of interest are indicated in those datasets rather than all co-existing mutations. The differences in *GADD45g* expression levels between *JAK2V617F*-positive and *JAK2V617F*-negative patients may be masked by the possible presence of other mutations that also have repressive effects on *GADD45g* expression. For the above reasons, we have deleted the description of the irrelevance of *Gadd45g* downregulation with somatic mutations in the Results section, all the statements and conclusions related to this result in the Introduction and Discussion sections, and Supplementary Fig.14A-D and its legend in the original manuscript.

Our previous work has shown that in addition to oncogenes, both promoter DNA methylation and histone deacetylation contributed to *GADD45g* silencing in AML (PMID: 33945602). To investigate whether DNA methylation is also involved in the decreased expression of *GADD45g* in MPNs, we treated MPN cell lines HEL and SET-2, and primary BMMNCs from MPNs patients with DNA demethylating agent Decitabine (2 μ M) for 96 hours. The mRNA level of *GADD45g* was upregulated by about twofold in these cells. Our methylation-specific PCR (MS-PCR) assay revealed that methylated *GADD45g* alleles were significantly decreased in HEL cell line while remained unchanged in SET-2 cell line upon Decitabine treatment. These data suggest that promoter DNA methylation plays a modest role in the silencing of *GADD45g* in MPNs, which is less effective than in AML.

To explore the contribution of histone modifications in *GADD45g* silencing in MPNs, we analyzed gene expression profilings of CD34⁺ BMMCs/PBMCs from MPN patients (GSE174060, GSE120362 and GSE103237), and found that *GADD45g* displays a markedly inverse correlation with *HDAC1* and *HDAC2* expression. Our previous study has shown that the selective HDAC1/2 inhibitor Romidepsin exhibits the strongest induction of *GADD45g* in AML (PMID: 33945602). We then treated MPN cell lines (3 nM) and BMMNCs from MPNs patients (5 nM) with Romidepsin for 48h, and observed that the mRNA levels of *GADD45g* were upregulated more than 4-fold in the cell lines, and approximately 3- to 4-fold in primary BMMNCs. ChIP-qPCR assay revealed that the levels of histones H3 and H4 acetylation at the *GADD45g* promoter were significantly upregulated upon Romidepsin treatment in both HEL and SET-2 cell lines.

In conclusion, some of the mechanisms underlying *GADD45g* silencing in MPN and AML are common, while others are distinct. *JAK2V617F* mutation and FLT3-ITD as well as MLL-AF9, oncogenic drivers in MPN and AML, respectively, exert an inhibitory effect on *GADD45g* expression. Histone deacetylation is the common epigenetic mechanism of *GADD45g* inactivation in MPN and AML, while the effect of DNA methylation is more pronounced in AML.

2. PDX studies in NSGS mice should be performed to determine the extent to which restoration of *GADD45g* restores MPN in the presence and/or absence of existing therapies such as JAK2 inhibitors and in MPNs with different mutations. These types of studies would help in better defining the role of *GADD45g* in MPN development. As presented, the link between MPN and *GADD45g* is tenuous at best.

Response: We appreciate the reviewer's suggestions. We had tried our utmost to generate PDX of MPN for years, but failed all the time, probably due to the poor engraftment potential of MPN disease-initiating cells (PMID: 32502268). Therefore, we used CDX model to determine the extent to which restoration of *GADD45g* restores MPN in the presence and/or absence of a JAK2 inhibitor. Briefly, Luciferase-expressing

HEL92.1.7 cells (HEL92.1.7-Luc) were transfected with Dox-inducible *GADD45g*-specific shRNA and engrafted into NSG mice by intravenous injection. The engraftment could be detected *in vivo* 10 days after transplantation, as indicated by the bioluminescence signal. The mice were then treated with either ruxolitinib or vehicle. Ruxolitinib administration resulted in a significant reduction in tumor burden, and a prominent elevation of *GADD45g* expression in hCD45-positive BMMNCs, which is consistent with our *in vitro* observations. Next, to determine whether downregulation of *GADD45g* had a restorative effect on disease burden, each of the ruxolitinib and vehicle treatment groups was subdivided into Dox and non-Dox treatment groups. The results revealed that, as expected, treatment with Dox resulted in an apparent increase in tumor burden and a significant decrease in survival compared with the vehicle. Furthermore, Dox-induced *GADD45g* knockdown with concurrent *JAK2* inhibition led to a partial restoration of tumor burden and a shortened survival, as compared to treatment with ruxolitinib alone. These observations indicate *GADD45g* partially mediates the pathogenicity of *JAK2V617F* in the MPN xenograft model.

The following content has been added to the revised manuscript with respect to the above two comments:

1) A new subsection entitled “*JAK2V617F* mutation and histone deacetylation are involved in *GADD45g* silencing in MPNs, and *GADD45g* downregulation partially mediates *JAK2V617F* activity in a MPN xenograft model” has been added to the Results section on pages 16-19 in the revised manuscript. This subsection includes the above results relevant to the effects of *JAK2V617F*, DNA methylation and histone deacetylation on *GADD45g*, with the CDX model inserted after *JAK2V617F*. The corresponding figures (Fig. 10a-p) and legends have been added. The statements “and *JAK2V617F* mutation and histone deacetylation contribute to its reduced expression.” and “The most common driver mutation *JAK2V617F* and histone deacetylation are involved in the *GADD45g* silencing in MPNs.” have been added in the Abstract and the last paragraph of Introduction, respectively.

2) A new discussion regarding the new findings on the repressive effect of *JAK2V617F*

on *GADD45g* expression and the role of *GADD45g* insufficiency in MPN has been added to the Discussion section on page 20, paragraph 3.

3) The methods relevant to the construction of *JAK2V617F* and wild-type *JAK2*, xenograft model, isolation of hCD45⁺ cells, MS-PCR and ChIP-qPCR assays, have been added to the Materials and methods section on page 27, paragraph 3, page 28, paragraph 4, page 26, paragraph 1 and page 30, paragraph 1-2, respectively. The drugs and antibodies used have been added to the Materials and methods section on page 25, paragraph 2 and page 31, paragraph 2, respectively.

Other comments:

1. Bars in Figure 1 look overlapping. The differences in engraftment appear modest. Engraftment analysis at 32 weeks should also be included. Have the authors looked at the homing of the knockout cells? This could help explain the modest difference in engraftment.

Response: We appreciate the reviewer's comment. We had performed the homing assay and no significant differences in the homing potential between My-HSCs from 6-month-old *Gadd45g*-insufficient mice and those from age-matched Ctrl group were observed. The self-renewal capacities of HSC are assessed by serial transplantations, and it would take more than one year to complete the analysis of engraftment at 32 weeks after the 4-month primary transplantation, far exceeded the time limit set by *Nature Communications* for revising the paper. Therefore, we cannot perform this experiment.

As we had described in the original manuscript, the hematological parameters of *Gadd45g*-deleted and Ctrl mice were examined every two months after birth. We had evaluated the self-renewal capacities of HSCs in mice at 4, 6 and 8 months of age, and the results of 8-month-old mice were not presented in the original manuscript due to the length and figure size limitations. The results revealed that the self-renewal capacities of My-HSCs from 8-month-old *Gadd45g*-deficient mice were prominently enhanced, and the extent of enhancement was more pronounced than those in mice

younger than 8 months. We have added this description to the Result section on page 7, paragraph 2, in the revised manuscript, as well as the corresponding Fig. 1k and legend. The results of self-renewal analysis of My-HSCs from 4-month-old mice and corresponding Figure 1l and legend in the original manuscript have been deleted because the serial competitive transplantations of LT-HSCs from 4-month-old mice had already been presented.

Since the homing potential of LT-HSCs from 4-month-old mice had been shown in the original manuscript, the negative results of homing from 6- and 8-month-old mice are not added to the revised manuscript, they are shown in the figure below (Figure 2A-B).

Figure 2 for Reviewer #2

Figure 2. *Gadd45g* deletions have no impact on the homing potential of My-HSCs from 6- and 8-month-old mice.

(A-B) Homing assay was performed by intravenous injection of two thousand freshly sorted My-HSCs from 6- (A) or 8-month-old (B) *Gadd45g*^{+/-} and *Gadd45g*^{-/-} mice or control mice into lethally irradiated recipients. Eighteen hours after transplant, BMMNCs were collected and CD45.2⁺ cells homing to recipient BM were analyzed by flow cytometry (n=5 mice per group).

Data are shown as means ± SD. ns, not significant (Student's *t* test).

2.As pointed out above, the mechanism of GADD45g repression in MPNs and is this mutation dependent should be explored further.

Response: Please see our responses to Major comment 1 and 2.

3.In PDX models of MPN does combination of AKT and JAK2 or RAC inhibitors correlate with amount of GADD45g repression in terms of disease burden and stem cell function *in vivo*.

Response: We appreciate the reviewer's comment. In order to assess whether combination of AKT and JAK2 or RAC inhibitors correlate with amount of *GADD45g* repression in MPN, we treated HEL cells with vehicle, JAK2 inhibitor (ruxolitinib, 500 nM) alone, JAK2 inhibitor in combination with AKT inhibitor (MK-2206, 10 μ M), JAK2 inhibitor in combination with RAC inhibitor (EHT 1864, 5 μ M), or AKT inhibitor in combination with RAC inhibitor. Cells were harvested at 24, 48 and 72h after treatment and the mRNA expression levels of *GADD45g* were analyzed by qRT-PCR. The results revealed that the combined application of JAK2 inhibitor with either AKT inhibitor or RAC inhibitor exerted no further effect on the *GADD45g* expression level, as compared to treatment with JAK2 inhibitor alone. Furthermore, the mRNA level of *GADD45g* was not altered upon dual inhibition of AKT and RAC, as compared to the vehicle group (Figure 3 for Reviewer #2). These data suggest that the combination of AKT and JAK2 or RAC inhibitors does not correlate with the amount of *GADD45g* repression in MPN cells. Given the negative *in vitro* results, we did not perform further *in vivo* experiments.

Figure 3 for Reviewer #2

Figure 3. The effects of different combinations of inhibitors on the expression levels of *GADD45g* in HEL cells.

HEL cells were treated with vehicle, ruxolitinib (Rux), Rux+MK-2206, Rux+EHT 1864 or MK-2206+EHT 1864, and the mRNA levels of *GADD45g* were examined by qRT-PCR at 24h (A), 48h (B), and 72h (C) after treatment.

Data are shown as means \pm SD. ***, $P < 0.001$; ns, not significant (Student's *t* test).

4. A recent study showed activation of Rac in Dnmt3a mutant cells and repression of AML/MPN like disease by Rac inhibitor in vivo; do Dnmt3a mutations repress *GADD45g*? if so, how?

Response: We appreciate the reviewer's comment. DNMT3A mutations are present in 5-10% of MPNs (PMID:32823933), the vast majority of which are loss-of-function mutations leading to aberrant hypomethylation and drives malignant progression via genetic de-repression (PMID: 28215704). However, *GADD45g* expression is repressed in MPNs, which is the opposite of the function of DNMT3A mutations. Therefore, it is unlikely that DNMT3A mutations repress the *GADD45g* expression in MPNs.

5. Does GADD45g bind to RAC2 but not RAC1 in primary patient derived MPNs? How is this specificity achieved? How does binding of the two or lack of result in increased Rac activation?

Response: We appreciate the reviewer's comment. Whether GADD45g binds to RAC2 but not RAC1 in primary cells from patients with MPNs cannot be assessed by co-IP or GST-pulldown assays due to insufficient cell numbers in each sample. As a fallback solution, we investigated the intracellular localization of the three proteins by immunofluorescence analysis in primary BMMNCs from patients with MPNs and cord blood CD34⁺ cells from healthy donors. The results revealed that both GADD45g and RAC2 exhibited a perinuclear distribution and areas of colocalization of them were clearly evident, consistent with that visualized in murine c-kit⁺ BM cells (original Figure 6B or revised Fig. 7b); whereas RAC1 displayed dispersed cellular locations with little perinuclear localization and its colocalization with GADD45g was low in cord blood CD34⁺ cells. Therefore, the distinct cellular localizations of RAC2 and RAC1 may be implicated in the specificity of binding. Our study had shown that the expression levels of *GADD45g* were significantly lower in BMMNCs from patients with MPNs than in those from healthy volunteers. Consistently, immunofluorescence analysis revealed that the BMMNCs derived from MPN patients expressed extremely low levels of GADD45g, and its colocalization with RAC2 was invisible or imperceptible. In the revised manuscript, the results of intracellular distribution of GADD45g, RAC2 and RAC1, and the colocalization of GADD45g and RAC2 in cord blood CD34⁺ cells from healthy donors have been added to the Results section on page 13, paragraph 3. The corresponding figure (Fig. 7c) and legend have been added. The antibodies used have been added to the Methods section on page 32, paragraph 3. Representative immunofluorescence micrographs showing intracellular localization of RAC1, RAC2 and GADD45g in the primary BMMNCs from MPN patients are not added to the revised manuscript, they are shown in the figure below (Figure 4 for Reviewer #2).

The present study revealed that binding of GADD45g and RAC2 leads to

inactivation of RAC2, however, there are no reports in the literature showing an association between GADD45g and RAC2. So, we suspected that other proteins might be involved. To identify the potential proteins, *GADD45g* was ectopically expressed in HEL cell line. Total cellular protein was extracted and GADD45g-interactome screening was performed using immunoprecipitation (IP) followed by mass spectrometry (MS). However, no regulators of RAC2 activity, e.g., GAPs, GDIs or GEFs, or proteins associated with them were found. Therefore, no further experiment can be performed due to the lack of further clues. The mechanisms by which the downregulation of *GADD45g* leads to activation of RAC2 remain unclear.

Figure 4 for Reviewer #2

Figure 4. Cellular localization of GADD45g, RAC2 and RAC1 in the primary BMMNCs from patients with MPN.

Representative immunofluorescence micrographs showing cellular distribution of GADD45g, RAC2 and RAC1 in the primary BMMNCs derived from MPNs patients. Panels represent nucleus (blue), GADD45g (green), RAC2 (orange), RAC1 (pink), and

merged images, respectively. Bar represents 5 μm .

Responses to Reviewer #3

Major concerns

1). Most of the results are based on *Gadd45* KO in the myeloid compartment which does not reflect the physiological levels of *Gadd45g* in HSC.

Response: We appreciate the reviewer's comment. We analyzed the gene expression profile of normal murine hematopoietic system in Bloodspot database, and observed that, in general, the expression level of *Gadd45g* was significantly higher in LT-HSCs than those in multiple distinct lineages of progenitors, while the mature cells exhibited the highest expression. Due to the length and figure size limitations, this analytical result is not added to the revised manuscript, and it is shown below for the reviewer.

Figure 1 for Reviewer #3

Figure 1. The expression patterns of *Gadd45g* in normal mouse hematopoietic developmental subsets.

The expression pattern of *Gadd45g* in the indicated cell populations in normal mouse hematopoietic system in Bloodspot database (GSE14833 and GSE6506). LT-HSC, long-term hematopoietic stem cell; ST-HSC, short-term hematopoietic stem cell; LMPP, lymphoid-primed multipotential progenitor; CLP, common lymphoid progenitor; GMP, granulocyte-monocyte progenitor; preGM, pre-granulocyte-macrophage progenitor; MkP, megakaryocyte progenitor; MkE, megakaryocyte erythroid progenitor; PreCFUE, pre-colony-forming unit erythroid; CFUE, colony-forming unit erythroid; ProE, proerythroblast.

Data are shown as means \pm SD. *, $P < 0.05$; **, $P < 0.01$ (Student's *t* test).

2). In a previous study (Blood Vol 138 #6 464-79, 2021) the authors show that GADD45g acts as a novel tumor suppressor in AML, where it exerts its effects via inhibition of E2F1 via the p38 MAPK–dependent signaling pathway and that silencing of GADD45g in AML is associated with epigenetic regulation and relevant leukemic oncogenes. The authors provide no data to explain this apparent dichotomy in *Gadd45g* signaling in HSC versus AML. This is puzzling and needs to be addressed.

Response: We appreciate the reviewer's comment. Our group has been working to elucidate the role of GADD45g in hematological diseases without involving normal HSC, so we think the reviewer is hoping we will explain the dichotomy in *Gadd45g* signaling in "MPN" versus AML, rather than "HSC versus AML". As mentioned by the reviewer, our previous work demonstrates that enforced expression of *GADD45g* exerts anti-tumor activities in AML via p38 MAPK-mediated E2F1 inhibition (Blood Vol 138 #6 464-79, 2021). The present study explores the pathogenic role of GADD45g insufficiency in myeloid malignancies, and shows that *Gadd45g* downregulation induces MPN in mice through a cascade of activation of RAC2, PAK1 and PI3K-AKT signaling pathways. We think it is rational that activation and inactivation of GADD45g have opposite effects on myeloid malignancies through different downstream signaling pathways.

Our previous study reveals that leukemic oncogenes FLT3-ITD and MLL-AF9,

which present with a high leukemic burden and confer a poor prognosis in patients with AML, as well as epigenetic mechanisms contribute to the silencing of *GADD45g* in AML (Blood Vol 138 #6 464-79, 2021). To elucidate the mechanisms underlying the reduction of *GADD45g* in MPN, in the revised manuscript, we first investigated whether *JAK2V617F*, the mutation critical in initiating and driving the development of MPNs is involved in *GADD45g* silencing in MPNs. In addition to its importance, the high mutation rate (95% of patients with PV) and the availability of inhibitors of *JAK2V617F* facilitate the study of its effects among the three driver mutations. We collected primary bone marrow specimens from patients with newly diagnosed PV (n=5), isolated bone marrow mononuclear cells (BMMNCs) and treated the cells with JAK2 inhibitor (ruxolitinib, Selleck, S1378, 50 nM) or vehicle for 24 hours. qRT-PCR analysis showed that the expression levels of *GADD45g* were markedly upregulated upon ruxolitinib treatment. We next transduced 293T cells with lentiviruses expressing *JAK2V617F*, wild-type *JAK2*, or empty vector constructs, and observed that only *JAK2V617F* overexpression resulted in a significant decrease in the expression level of *GADD45g*. These data suggest that *GADD45g* expression is repressed by *JAK2V617F* mutation in MPNs. As for the other two driver mutations, the mutation rate of *MPL* is very low (PMID: 16868251), while *CALR* is a loss-of-function mutation that requires overexpression for functional assessment (PMID: 32733014). However, primary MPN cells cannot tolerate the cytotoxic effects of lentiviral-mediated gene overexpression. Therefore, whether *MPL* and *CALR* mutations are associated with the aberrant expression of *GADD45g* are not assessed in this study.

The inhibitory effect of *JAK2V617F* mutation on *GADD45g* expression, as indicated by our observations, is not consistent with the analytical results from the public datasets presented in our original manuscript, which concluded that the reduced expression of *GADD45g* was irrelevant to somatic mutations (Supplementary Fig.14A-D). A possible explanation is that only mutations of interest are indicated in those datasets rather than all co-existing mutations. The differences in *GADD45g* expression levels between *JAK2V617F*-positive and *JAK2V617F*-negative patients may be masked by the possible presence of other mutations that also have repressive effects on

GADD45g expression. Therefore, we have deleted the description of the irrelevance of *Gadd45g* downregulation with somatic mutations in the Results section, all the statements and conclusions related to this result in the Introduction and Discussion sections, and Supplementary Fig.14A-D and its legend in the original manuscript.

Our previous work has shown that in addition to oncogenes, both promoter DNA methylation and histone deacetylation contributed to *GADD45g* silencing in AML (Blood Vol 138 #6 464-79, 2021). To investigate whether DNA methylation is also involved in the decreased expression of *GADD45g* in MPNs, we treated MPN cell lines HEL and SET-2, and primary BMMNCs from MPNs patients with DNA demethylating agent Decitabine (2 μ M) for 96 hours. The mRNA level of *GADD45g* was upregulated by about twofold in these cells. Our methylation-specific PCR (MS-PCR) assay revealed that methylated *GADD45g* alleles were significantly decreased in HEL cell line while remained unchanged in SET-2 cell line upon Decitabine treatment. These data suggest that promoter DNA methylation plays a modest role in the silencing of *GADD45g* in MPNs, which is less effective than in AML.

To explore the contribution of histone modifications in *GADD45g* silencing in MPNs, we analyzed gene expression profilings of CD34⁺ BMMCs/PBMCs from MPN patients (GSE174060, GSE120362 and GSE103237), and found that that *GADD45g* displays a markedly inverse correlation with *HDAC1* and *HDAC2* expression. Our previous study has shown that the selective HDAC1/2 inhibitor Romidepsin exhibits the strongest induction of *GADD45g* in AML (Blood Vol 138 #6 464-79, 2021). We then treated MPN cell lines (3 nM) and BMMNCs from MPNs patients (5 nM) with Romidepsin for 48h, and observed that the mRNA levels of *GADD45g* were upregulated more than 4-fold in the cell lines, and approximately 3- to 4-fold in primary BMMNCs. ChIP-qPCR assay revealed that the levels of histones H3 and H4 acetylation at the *GADD45g* promoter were significantly upregulated upon Romidepsin treatment in both HEL and SET-2 cell lines.

In conclusion, some of the mechanisms underlying *GADD45g* silencing in MPN

and AML are common, while others are distinct. *JAK2V617F* mutation and FLT3-ITD as well as MLL-AF9, oncogenic drivers in MPN and AML, respectively, exert an inhibitory effect on *GADD45g* expression. Histone deacetylation is the common epigenetic mechanism of *GADD45g* inactivation in MPN and AML, while the effect of DNA methylation is more pronounced in AML.

The following content has been added to the revised manuscript, with an *in vivo* study using a xenograft model inserted after the effect of *JAK2V617F* (For details on the *in vivo* study, please see the responses to Reviewer #2's Major comments 1, 2):

1) The above results relevant to the effects of *JAK2V617F*, DNA methylation and histone deacetylation on *GADD45g* have been added to the Results section on pages 16-19, in the revised manuscript. The corresponding figures (Fig.10a-p) and legends have been added. The statements “and *JAK2V617F* mutation and histone deacetylation contribute to its reduced expression.” and “The most common driver mutation *JAK2V617F* and histone deacetylation are involved in the *GADD45g* silencing in MPNs.” have been added in the Abstract and the last paragraph of Introduction, respectively.

2) A new discussion regarding the new findings on the repressive effect of *JAK2V617F* on *GADD45g* expression and the role of *GADD45g* insufficiency in MPN has been added to the Discussion section on page 20, paragraph 3.

3) The methods relevant to the construction of *JAK2V617F* and wild-type *JAK2*, MS-PCR and ChIP-qPCR assays have been added to the Materials and methods section on page 27, paragraph 3 and page 30, paragraph 1-2, respectively. The drugs and antibodies used have been added to the Materials and methods section on page 25, paragraph 2 and page 31, paragraph 2, respectively.

REVIEWERS' COMMENTS

Reviewer #1 (Remarks to the Author):

The responses to reviewers and changes are noted. most of the minor changes have been adequately addressed however there are a number of outstanding issues.

1. Inclusion of data showing down regulation of GADD45G. These are critical to the hypothesis and provide the rationale for why the authors have commenced this research. Rather than "to validate the mouse results" surely the purpose is to start with human data and then see whether the mouse data validate the human results. I still think Figure 4a-D should be the first figures of the paper and then follow from there, but accept that this is an editorial decision.

2. The new data 1k can not be concluded to show that 8 month mice have a greater difference than 4 months (1a) because these are presumably different experiments without head to head comparisons.

3. Figure 3 should be relegated to supplemental data. this adds very little to primary data and is essentially negative data.

4. Erythroid maturation - no explanation is given.

5. There were many suggestions to include important data into primary figures. The explanation is that the figures are already too large. I would suggest that representative data or data that repeat primary findings (e.g. 4j, 4k, figure 3, 1k, 1i, 1j etc.) could be moved to supplementary to allow the inclusion of important data in primary figures. I don't accept that the figures are already too large.

Reviewer #2 (Remarks to the Author):

although authors have tried to address some of my concerns, a significant number of them have not been addressed. In particular, studies in primary MPN have not been performed. Additionally, a significant amount of overlap of the current study appears with their recent BLOOD paper on AML. Thus, the lack of novelty of the work continues to be an additional major concern.

Reviewer #3 (Remarks to the Author):

The revised manuscript is much improved and suitable for publication.

RESPONSE TO REVIEWERS' COMMENTS

Responses to Reviewer #1

Specific comments

1. Inclusion of data showing down regulation of GADD45G. These are critical to the hypothesis and provide the rationale for why the authors have commenced this research. Rather than “to validate the mouse results” surely the purpose is to start with human data and then see whether the mouse data validate the human results. I still think Figure 4a-d should be the first figures of the paper and then follow from there, but accept that this is an editorial decision.

Response: We agree with the reviewer's comment. As suggested by the reviewer, we have moved the human data to the beginning of the results description in the Abstract, Introduction, and Results sections of the revised manuscript. The reason for starting the research has also been changed to “However, whether GADD45g plays a role in the development of MPN remains unknown” on page 5, paragraph 2 in the Introduction section. The original Figure 4 and Supplementary Fig. 8, which describes the human results, have been renumbered as Figure 1 and Supplementary Fig. 1, respectively. The original figure numbers associated with these revisions have been changed accordingly.

2. The new data 1k cannot be concluded to show that 8 month mice have a greater difference than 4 months (1a) because these are presumably different experiments without head to head comparisons.

Response: We agree with the reviewer's comment. The original “Furthermore, the enhancement of the self-renewal capacities of My-HSCs was more pronounced in 8-month-old *Gadd45g*^{+/-} and *Gadd45g*^{-/-} mice than those in mice younger than 8 months (Fig. 1k).” has been changed to “Furthermore, the My-biased HSCs from 8-month-old *Gadd45g*-insufficient mice also exhibited

significantly enhanced self-renewal, as compared to age-matched Ctrl mice (Fig. 2q).” in the Results section on page 8, paragraph 2 of the revised manuscript.

3. Figure 3 should be relegated to supplemental data. this adds very little to primary data and is essentially negative data.

Response: As suggested by the reviewer, the original Fig.3 has been moved to supplementary data and renumbered as Supplementary Fig. 7 in the revised manuscript.

4. Erythroid maturation - no explanation is given.

Response: We appreciate the reviewer’s comment. Our results show that the numbers of megakaryocyte-erythroid progenitors (MEPs) and Ter119⁺ cells were significantly reduced in the bone marrow of our MPN-like mouse model, explaining the anemia detected. In addition, the proportions of MEPs and Ter119⁺ cells were significantly augmented in the spleens of the moribund mice. Such phenotypes are not uncommon in mouse models of MPN (PMID:23221337, PMID:28232469, PMID:26834157). Extramedullary erythropoiesis usually occurs as a compensatory phenomenon in response to insufficient bone marrow blood cell production, but cannot ameliorate anemia (PMID: 23221337, PMID: 2757009). Therefore, the expansion of red cell precursors in the spleen is not contradictory with the anemic phenotype.

5. There were many suggestions to include important data into primary figures. The explanation is that the figures are already too large. I would suggest that representative data or data that repeat primary findings (e.g. 4j, 4k, figure 3, 1k, 1i, 1j etc.) could be moved to supplementary to allow the inclusion of important data in primary figures. I don’t accept that the figures are already too large.

Response: We appreciate the reviewer's comment. In accordance with the many suggestions of the reviewer in this and the last revised version "All of the analysis of primary mice has been relegated to supplementary data." and "MPN-like disease is transplantable, and rescued by re-expression of *GADD45g*. These data are important and should be shown in primary data with figure 2.", the following changes have been made in the revised manuscript:

- 1) The entire original Fig. 4 which describes the human data, has been renumbered as Fig.1.
- 2) To display the most important hematological parameters of the BM and spleen from 6-month-old mice in the main figure, the original Fig. 1a-h, 1k, Supplementary Fig. 4b-g, Supplementary Fig. 5a-b, and Supplementary Fig. 5h, have been combined and renumbered as Fig. 2a-q.
- 3) The original Fig. 2a-n has been renumbered as Fig. 3a-n.
- 4) The original Fig. 3, the negative results of WES, has been moved to supplementary data and renumbered as Supplementary Fig. 7.
- 5) The original Fig. 2o-p and original Supplementary Fig. 7a-i, showing that the MPN-like disease is transplantable and rescued by re-expression of *Gadd45g*, have been combined and renumbered as Fig. 4a-k.
- 6) The original Supplementary Fig. 4a, Supplementary Fig. 5c-g and original Fig. 1i-j, which show the less important hematological parameters of the PB and spleen from 6-month-old *Gadd45g*-deficient mice and the myeloid-biased differentiation in the BM after transplantation, have been combined and renumbered as Supplementary Fig. 5a-h.

Responses to Reviewer #2

Although authors have tried to address some of my concerns, a significant number of them have not been addressed. In particular, studies in primary MPN have not been performed. Additionally, a significant amount of overlap of the current study appears with their recent BLOOD paper on AML. Thus, the lack of novelty of the work continues to be an additional major concern.

Response: We appreciate the reviewer's comment. We are sorry that the PDX model is not available in our study. We had asked many colleagues in different institutes, universities and hospitals, but none of them had the model, and they all agree that the model is difficult to establish due to the poor engraftment potential of MPN disease-initiating cells. Therefore, to address the reviewer's concerns, we used the CDX model, which is widely used and has proven valuable in the cancer field.

Our BLOOD paper explores the anti-leukemic effects of GADD45g activation in AML, while the present manuscript investigates the pathogenic role of GADD45g insufficiency in MPNs. There is little overlap between them. Therefore, we believe that the novelty of this work is not diminished by the previously published one.